

# Breaking epigenetic shackles: targeting ARID1A methylation and the PI3K/AKT/mTOR-PD-L1 axis to overcome immune escape in gastric cancer

Xueqin Duan[1,2,*], Xingfa Huo[1,2,*], Yuming Zhang[3], Hongwei Lan[1,2], Fangfang Yang[1,2], Xiaochun Zhang[1] and Na Zhou[1]

[1] Precision Medicine Center of Oncology, The Affiliated Hospital of Qingdao University, Qingdao, Shandong, China
[2] Department of Medicine, Qingdao University, Qingdao, Shandong, China
[3] Department of Oncology, Qingdao Central Hospital of Health and Rehabilitation University, Qingdao, Shandong, China
[*] These authors contributed equally to this work.

## ABSTRACT

**Objective**. AT-rich interaction domain 1A (ARID1A), is frequently mutated in cancer, leading to loss-of-function and posing challenges to therapeutic targeting. This study aimed to systematically explore epigenetic regulation of ARID1A, specifically promoter hypermethylation, in gastric cancer (GC) and its functional/immunological consequences.

**Methods**. We employed multi-omics bioinformatics analyses (UALCAN, cBioPortal, MEXPRESS and UCSC Xena) combined with *in vitro* functional validation in GC cell lines, including pharmacological demethylation using 5-Aza-2'-deoxycytidine (5-aza-CdR) and mechanistic interrogation *via* AKT agonism (SC79).

**Results**. Promoter hypermethylation was identified as a key mechanism silencing ARID1A transcriptional, showing a significant negative correlation between methylation $\beta$-values and mRNA expression (Spearman's $\rho = -0.29$, $p = 2.06 \times 10^{-8}$). 5-aza-CdR treatment restored ARID1A expression ($p < 0.001$), suppressed malignant phenotypes (proliferation, invasion, and apoptosis resistance), and revealed that ARID1A lose activates the phosphatidylinositol 3-kinase (PI3K)/protein kinase B (AKT)/mammalian target of rapamycin (mTOR) pathway (elevated p-AKT, p-mTOR) and upregulates PD-L1. Rescue experiments with SC79 reversed 5-aza-CdR's effects, confirming the ARID1A-PI3K/AKT/mTOR-PD-L1 axis. Integrative analysis linked ARID1A hypermethylation to elevated immune/ESTIMATE scores ($p < 0.05$).

**Conclusion**. ARID1A promoter hypermethylation drives an epigenetic-immune checkpoint cascade in GC. Combined with its association with immune signatures and PD-L1 upregulation, ARID1A hypermethylation emerges as a candidate biomarker for predicting immune checkpoint blockade (ICB) responsiveness and patient stratification in GC. Future studies should evaluate 5-aza-CdR-ICB-AKT inhibitor regimens in advanced models to guide clinical translation.

Corresponding authors
Xiaochun Zhang,
zxc9670@qdu.edu.cn
Na Zhou, zhouna@qdu.edu.cn

## INTRODUCTION

Gastric cancer (GC) is the fifth most prevalent malignancy globally and the fourth leading cause of cancer-related mortality (*Sung et al., 2021*). Despite therapeutic advances, the prognosis of GC remains poor, with a 5-year survival rate below 30% (*Allemani et al., 2018*). Post-resection adjuvant chemotherapy or chemoradiation is standard for operable patients (*Li, Doherty & Wang, 2022*); however, approximately 60% of patients experience recurrence or metastasis (*Ajani, 2005*). The immune checkpoint blockade (ICB) has transformed oncology by inducing durable responses in refractory metastatic cancers (*Lebedev et al., 2023*). However, its clinical utility is limited due to a low objective response rates of approximately 20% (*Doroshow et al., 2021*) and unreliable biomarkers. PD-L1 expression is a common biomarker for stratifying patients eligible for immune checkpoint blockade (ICB). However, its reliability is limited because it is dynamically regulated by the tumor microenvironment (TME) and is subject to technical variability in detection. Therefore, more robust predictive biomarkers are needed. Gastric carcinogenesis involves environmental and genetic factors. *Helicobacter pylori* infection drives epigenetic dysregulation *via* methyltransferase suppression (*e.g.*, METTL14) (*Cui et al., 2025*) and aberrant DNA methylation (*Gu et al., 2025*). Tobacco, dietary carcinogens, and radiation further induce genome-wide hypermethylation (*Loe, Zhu & Kim, 2023*). These alterations start early (*Li et al., 2022*), can be reversed (*Feng & De Carvalho, 2022*), and cause promoter hypermethylation to silence tumor suppressors (*Gu et al., 2025*). This epigenetic vulnerability underscores the therapeutic relevance of methylation-silenced tumor suppressors like ARID1A.

AT-rich interaction domain 1A (ARIDA), which encodes the BAF250a subunit of the Switch/Sucrose Non-Fermentable (SWI/SNF) chromatin remodeling complex (*Mandal et al., 2022*), is recurrently mutated across cancers, including ovarian clear cell (*Jones et al., 2010*), endometrial (*Wiegand et al., 2010*), and gastric carcinomas (*Wang et al., 2011*). Most ARID1A mutations are inactivating and drive protein loss (*Mandal et al., 2022*), with mutation frequencies varying among GC subtypes (*e.g.*, elevated in Epstein-Barr virus (EBV)-positive GC) (*Cancer Genome Atlas Research Network, 2014*). Consequently, ARID1A mutations are not ideal therapeutic targets. However, recent evidence suggests that epigenetic silencing *via* promoter hypermethylation may similarly disrupt the tumor-suppressive function of ARID1A (*Li et al., 2024*). DNA methylation—the addition of methyl groups to cytosine residues within CpG islands—is a key epigenetic regulator of gene expression and tumorigenesis (*Kuang et al., 2023*; *Loe, Zhu & Kim, 2023*). Although ARID1A promoter hypermethylation has been implicated in squamous cell carcinoma (*Luo et al., 2020*), its role in GC and therapeutic implications remain unclear.

Notably, ARID1A deficiency reshapes the TME and modulates immunotherapy responses. Preclinical models have demonstrated that ARID1A-mutated ovarian tumors
exhibit PD-L1 upregulation (*Shen et al., 2018*), suggesting that its mutation status may predict ICB efficacy. However, whether ARID1A epigenetic silencing similarly drives PD-L1 dysregulation in GC remains unexplored.

In this study, we investigate the epigenetic regulation of ARID1A in GC progression. Through integrated multi-omics analyses and functional validation, we uncover ARID1A promoter hypermethylation as a mechanism of transcriptional silencing, with inversely correlated with mRNA expression. *In vitro* studies demonstrate that ARID1A epigenetic inactivation promotes malignant phenotypes *via* PI3K/AKT/mTOR pathway activation, concomitant with PD-L1 upregulation. Pharmacological demethylation (5-aza-CdR) reverses this oncogenic cascade, and rescue experiments with an AKT agonist confirm the functional reversibility of this axis. These findings suggest a DNA methylation-driven ARID1A-PI3K/AKT/mTOR -PD-L1 cascade as a potential immune evasion pathway, providing preclinical rationale for exploring epigenetic modulation to enhance immunotherapy efficacy in GC.

## MATERIALS AND METHODS

### Methylation analysis of ARID1A

The methylation status of ARID1A in GC tissues and adjacent normal tissues was evaluated using the UALCAN database (http://ualcan.path.uab.edu/analysis.html, accessed 15 September 2024). The correlation between ARID1A methylation density and mRNA expression was evaluated using Spearman correlation analysis. The data for this analysis were downloaded from cBioPortal (http://www.cbioportal.org, Stomach Adenocarcinoma dataset, accessed 20 September 2024). Additional validation of site-specific methylation effects was performed using MEXPRESS (https://mexpress.ugent.be/, accessed 20 September 2024). To validate site-specific methylation patterns identified in the MEXPRESS analysis (FDR-adjusted *p*-value < 0.05), RNA expression profiles, methylation data, and clinical information for GC were retrieved from the UCSC Xena (accessed September 2024). Publicly available datasets included RNA-seq expression profiles from 375 GC and 33 normal tissues and methylation data from 396 tumors and two normal tissues (Illumina Human Methylation 450K BeadChip). Gene identifiers were converted to gene symbols using Ensembl (Homo sapiens, http://asia.ensembl.org/index.html, accessed 20 September 2024). All data were obtained from publicly accessible, de-identified databases (UALCAN, cBioPortal, MEXPRESS, UCSC Xena). According to NCI Guidelines (2015) and TCGA policies (https://www.cancer.gov/ccg/research/genome-sequencing/tcga/history/ethics-policies), this secondary analysis of pre-deidentified public data is exempt from ethics committee approval.

### Identification of differentially expressed genes

Gastric cancer patients were stratified by ARID1A promoter $\beta$-values (cg05445839) into hypermethylated (Hyper-group, ≥75th percentile) and hypomethylated (Hypo-group, ≤25th percentile) subgroups. Methylation-regulated differentially expressed genes (MeDEGs) were identified using the DESeq2 package (version 1.38.3) in R software,

with significant thresholds set at $|\log_2 \text{FoldChange}| \geq 1.5$ and false discovery rate (FDR) adjusted $p$-value < 0.05 for subsequent analyses.

## Functional enrichment analysis and protein interaction network analysis

MeDEGs were annotated using Gene Ontology (GO) functional enrichment and Kyoto Encyclopedia of Genes and Genomes (KEGG) pathway analyses. GO enrichment included biological processes (BP), molecular functions (MF), and cellular components (CC) (*Ashburner et al., 2000*). KEGG, a database resource, was used to elucidate gene functions at the molecular and higher-order levels, including biochemical pathways (*Kanehisa et al., 2017*). Enrichment analysis was conducted using a hypergeometric test. For gene set functional enrichment, the clusterProfiler R package (v3.14.3) (*Yu et al., 2012*) was used with GO annotations serving as the background reference to map genes to predefined functional categories. Statistically significant terms were defined by a $p$-value < 0.05 and a false discovery rate (FDR) <0.05. Protein-protein interaction (PPI) networks were constructed using the STRING (v11.5, confidence score >0.7, active sources: Experiments & Databases) (*Szklarczyk et al., 2019*) and visualized with Cytoscap (v3.10.0) (*Shannon et al., 2003*). Hub genes were identified using the cytohubba plugin (v0.1) in Cytoscape (v3.10.0) by selecting the top 10 nodes ranked by Closeness centrality. Functional modules were detected with MCODE (v2.0.2) using default parameters (degree cutoff = 2, node score cutoff = 0.2, k-core = 2, max depth = 100).

## Gene set enrichment analysis

Gene set enrichment analysis (GSEA) was conducted using the clusterProfiler (v3.14.3) in R to identify significant differences in predefined gene sets between the methylation-defined groups, applying Benjamini–Hochberg correction (FDR-adjusted $p$-value < 0.05) and size filtering (10–500 genes). Statistically significant enrichment was defined by the following cutoff criteria: |Normalized Enrichment Score (NES)| > 1, nominal $p$- value < 0.05, and FDR-adjusted $p$-value < 0.05.

## TISIDB database analysis

The "Chemokine" module of the TISIDB database (http://cis.hku.hk/TISIDB/, accessed 1 October 2024) was used to investigate the association between ARID1A and chemokine/chemokine receptor expression levels in tumor-infiltrating immune cells. To further elucidate the immunological relevance of ARID1A in cancer, the "Immunomodulator" module was used to evaluate correlations between ARID1A expression and immune checkpoint gene levels.

## Comprehensive immune landscape profiling in methylation-defined subgroups

Immunological heterogeneity between the hypermethylated and hypomethylated subgroups was comprehensively profiled using seven computational algorithms and visualized *via* hierarchical clustering heatmaps (Euclidean distance; ComplexHeatmap v2.14.0). Immune cell composition was first deconvoluted using CIBERSORT to calculate

the relative enrichment scores for 22 functionally distinct immune cell subtypes (*Newman et al., 2015*). TME characteristics, including the immune/stromal cell ratios and neoplastic purity, were quantified using the ESTIMATE algorithm (*Yoshihara et al., 2013*). Stromal-immune interplay was further resolved using an MCP-counter, which provides absolute quantification of eight stromal and two immune cell populations (*Becht et al., 2016*). The EPIC framework was implemented to differentiate cancer epithelial cells from seven non-malignant microenvironment components, enhancing the resolution of tumor-immune interactions (*Racle et al., 2017*). Bulk transcriptome data were analyzed *via* TIMER 2.0 to estimate the abundance of six functionally key immune infiltrates ($CD8^+$ T cells, B cells, macrophages, *etc.*) (*Li et al., 2020*). The quanTIseq pipeline enabled absolute quantification of 10 immune cell types through signature matrix deconvolution of RNA-seq data (*Finotello et al., 2019*). Finally, the Immunophenoscore (IPS) algorithm integrated MHC class I/II presentation, immunostimulatory/checkpoint molecules, and effector cell signatures to generate composite immunogenicity indices (*Charoentong et al., 2017*).

## Cell culture and reagent preparation

Human GC cell lines HGC-27 and AGS were cultured in RPMI 1640/F-12 medium (Gibco, Thermo Fisher Scientific, Waltham, MA, USA) supplemented with 10% fetal bovine serum (FBS; Gibco, Waltham, MA, USA) and 1% penicillin/streptomycin (Beyotime, Jiangsu, China). All cells were maintained in a humidified incubator at 37 °C with 5% $CO_2$. Cell line authenticity was routinely verified by short tandem repeat DNA profiling, and mycoplasma contamination was excluded using the MycoBlue Mycoplasma Detector (Vazyme Biotech, Nanjing, China) before experimental use. 5-Aza-2'-deoxycytidine (5-aza-CdR; Selleck Chemicals, Houston, TX, USA) was reconstituted in dimethyl sulfoxide (DMSO; Gibco, Waltham, MA, USA) to a stock concentration of 50 mM, aliquoted, and stored at −80 °C until use. SC79 (MedChemExpress, Monmouth Junction, NJ, USA) was similarly prepared in DMSO at a stock concentration of 20 mM and stored under identical conditions.

## RNA isolation and quantitative real-time PCR analysis

Total RNA was isolated using the RNA-easy Isolation Reagent Kit (Vazyme Biotech, Nanjing, China) according to the manufacturer's protocol. RNA concentration and purity were measured spectrophotometrically using a NanoDrop 2000 system (Thermo Fisher Scientific, Waltham, MA, USA). For cDNA synthesis, 1 μg of total RNA was reverse-transcribed using SuperScript™ IV Reverse Transcriptase (200 U/μL; Invitrogen, Waltham, MA, USA) under conditions specified by the supplier.

Quantitative real-time PCR (qRT-PCR) was performed on a Bio-Rad CFX96 Touch Real-Time PCR Detection System using FASTStart SYBR Green Master Mix (PeqLab Biotechnologies GmbH). Each 20 μL reaction contained 2 μL of cDNA template. Primer sequences were as follows: ARID1A: Forward: 5'-CAGTACCTGCCTCGCACATA-3'; Reverse: 5'-GCCAGGAGACCAGACTTGAG-3'.

$\beta$-actin (ACTB; endogenous control): Forward: 5'-TCCTGTGGCATCCACGAAACT-3'; Reverse: 5'-GAAGCATTTGCGGTGGACGAT-3.' Thermal cycling conditions included an initial denaturation at 95 °C for 30 s, followed by 40 cycles at 95 °C for 10 s and 60 °C for

30 s. Relative gene expression was calculated using the $2^{\wedge}(-\Delta\Delta Ct)$ method, with $\beta$-actin as the normalization control.

## DNA isolation and quality assessment

Genomic DNA was isolated from GC cell lines using the TIANamp Genomic DNA Kit according to the manufacturer's protocol. DNA concentration and purity were assessed spectrophotometrically using a NanoDrop™ 2000 system (Thermo Fisher Scientific, Waltham, MA, USA). Samples meeting the following quality criteria were retained for downstream analyses: DNA concentration $\geq$100 ng/µL and A260/280 absorbance ratios approximating 1.8 ($\pm$0.2).

## Bisulfite treatment

Genomic DNA (1 µg per sample) was subjected to bisulfite conversion using a DNA Bisulfite Conversion Kit (Beyotime Biotechnology, Jiangsu, China) according to the manufacturer's protocol. The conversion reaction was performed in a thermal cycler under the following conditions: initial denaturation at 95 °C for 3 min, followed by 12 cycles at 95 °C for 30 s and 70 °C for 10 min, with a final hold at 4 °C. The converted DNA was then transferred to purification columns for desulfonation, washing, and elution. The bisulfite-converted DNA samples were ultimately stored at −20 °C for further molecular analyses.

## Methylation-specific PCR analysis of ARID1A

The bisulfite-converted DNA was subjected to methylation-specific PCR (MSP). The MSP primers were designed using the MethPrimer program (MethPrimer-Design MSP/BSP primers and CpG islands prediction; Li Lab, PUMCH; available at http://www.urogene.org, accessed 5 December 2024). The following primer sets were used for the MSP analysis:

ARID1A-M-F: 5'-GTAATAATTTGGCGTTTTAGCGA-3';
ARID1A-M-R: 5'-ACCCAATCCTTATAAAAAAAATCGT-3' (methylated-specific),
ARID1A-U-F: 5'-GGGGTAATAATTTGGTGTTTTAGTG-3',
ARID1A-U-R: 5'-CCCAATCCTTATAAAAAAAATCATC-3' (unmethylated-specific).

The 50 uL MSP-PCR reaction mixture contained 5 µL of 10× PCR Buffer ($Mg^{2+}$ Plus, 25 mM; R007A, TaKaRa Bio Inc., Shiga, Japan), 4 µL of dNTP Mixture (2.5 mM, R007A, TaKaRa, Shiga, Japan), 1 µL each of forward and reverse ARID1A primers (100 µM, Sangon Biotech, Shanghai, China), 0.25 µL of TaKaRa LA Taq HS DNA polymerase (5U/uL, R007A, TaKaRa, Shiga, Japan), 36.75 µL of nuclease-free $H_2O$, and 2 µL of bisulfite-converted DNA template (50 ng/µL). Thermal cycling conditions for MSP amplification were as follows: Initial denaturation at 95 °C for 5 min, followed by 40 cycles of denaturation at 95 °C for 30 s, primer-specific annealing (59 °C for methylated [M] primers or 55 °C for unmethylated [U] primers) for 45 s, and extension at 72 °C for 45 s, with a final extension at 72 °C for 10 min. Methylation-specific PCR products were resolved on a 2% agarose gel pre-stained with YeaRed Nucleic Acid Gel Stain (YEASEN Biotechnology, Cat# 10202ES76, Shanghai, China) at 140 V for 40 min. DNA bands were visualized under UV light, and the 106 bp target fragments were identified by comparison with DL2000 Plus DNA Marker (MD101-01-AA; 100–5000 bp; Vazyme, Beijing, China).

## Lentiviral transduction for ARID1A knockdown and cell infection

Lentiviral vectors targeting ARID1A (shARID1A:5'-TTCTCCGAACGTGTCACGT-3') and non-targeting control shRNA (shCtrl: 5'-GTTGATGAACTCATTGGTT-3') were obtained from Genepharma (China). For lentiviral transduction, GC cells at 40% confluence were co-transduced with two lentiviral constructs: one carrying a ARID1A-specific shRNA and the other containing scrambled control shRNA. Transduction was performed at a multiplicity of infection of 10 in serum-free basal medium to enhance viral entry efficiency. After 8–12 h of incubation, the medium was replaced with a complete growth medium containing 10% FBS and maintained for 72 h post-transduction.

GFP expression in transduced cells was quantitatively assessed using an A1 confocal laser-scanning microscope (Nikon, Japan). Successful transduction was confirmed by the detection of green fluorescence indicative of GFP-positive cells. Subsequently, transduced cells underwent rigorous selection with 2 μg/mL puromycin for 7 d to isolate puromycin-resistant clones. Validation of stable ARID1A knockdown was confirmed by western blotting and qRT-PCR before downstream functional assays.

## Western blot

Cells were washed with phosphate-buffered saline (PBS) and lysed in RIPA lysis buffer (P0013B, Beyotime) supplemented with Protease Inhibitor Cocktail (HY-K0010, MedChemExpress) and phosphatase inhibitors (HY-K0021, MedChemExpress). Lysates were incubated on ice for 30 min, centrifuged at 12,000 rpm for 10 min at 4 °C, and supernatants were collected. Protein concentrations were quantified using the BCA Protein Assay Kit (P0010S, Beyotime). Equal amounts of protein from each sample were separated by sodium dodecyl sulfate-polyacrylamide gel electrophoresis (SDS-PAGE) and transferred onto nitrocellulose membranes (EMD Millipore, Billerica, MA, USA). Membranes were blocked with 5% non-fat milk at room temperature and then incubated overnight at 4 °C with primary antibodies, including: Anti-ARID1A (1:1000, ab182560, Abcam), Anti-PD-L1 (1:1000, PB0166, BOSRER), Anti-PI3K (1:2000, 60225-1-AP, Proteintech), Anti-p-PI3K (1:1000, #4228, Cell Signaling Technology [CST]), Anti-AKT (1:1,000, D290056, Sangon), Anti-p-AKT (Ser473) (1:1,000, T40067, Abmart), Anti-mTOR (1:1000, #CBD-13, BOSRER), Anti-p-mTOR (1:1000, #IFF-13, BOSRER), and Anti-β-actin (1:10,000, A3854, Sigma-Aldrich), Tubulin (1:2000,66200-1-AP, Proteintech). Membranes were subsequently incubated with horseradish peroxidase-conjugated goat anti-mouse/rabbit IgG secondary antibodies (LEF22060, Life-iLab) at room temperature. Protein bands were visualized *via* chemiluminescence, and signal intensities were quantified by densitometry using the ImageJ software.

## Migration and invasion assays

For migration analysis, GC cells ($3 \times 10^4$ cells/well), either treated or untreated, were seeded into the upper chamber of Transwell inserts (8-μm pore size, Corning Inc., Corning, NY, USA). The lower chamber was filled with 600 μL of complete medium alone or medium containing a specified concentration of 5-aza-CdR. After 24 h of incubation, cells in the upper chamber were fixed with 4% paraformaldehyde, stained with 0.1% crystal violet,

and imaged under an optical microscope. The number of migrating cells was quantified by counting cells in six randomly selected microscopic fields per insert. For the invasion assay, the protocol was identical, except that Transwell inserts were pre-coated with 4 μL of Matrigel (Corning Inc., Corning, NY, USA) before cell seeding.

## Wound-healing assays

A wound healing assay was performed to evaluate its effect on the cellular migratory capacity. Following successful transfection, well-grown cells were seeded into six-well plates and cultured to form confluent monolayers. A sterile 200 μl pipette tip was used to create a linear mechanical wound by vertically scraping the cellular monolayer. After three consecutive washes with PBS to eliminate cellular debris, a serum-free medium was added. Wound closure progression was systematically monitored at baseline 0 h and 24 h intervals, with photomicrographic documentation performed under standardized conditions. High-resolution images were captured using an Olympus IX83 inverted light microscope equipped with phase-contrast optics. Quantitative analysis of cellular migration was conducted by measuring the interwound distance using the ImageJ software (National Institutes of Health, Bethesda, MD, USA) with calibrated digital image-processing protocols.

## Cell counting kit-8 cell proliferation assays

Cell viability was systematically evaluated using the Cell Counting Kit-8 (CCK-8; Sangon Biotech, Shanghai, China) following the manufacturer's recommended protocols. GC cells were seeded into 96-well microplates at a density of 1,000 cells per well and allowed to stabilize overnight. After designated incubation intervals, 10 μl of CCK-8 reagent was aseptically dispensed into each experimental well. The plates were then transferred to a light-protected incubator and maintained at 37 °C for 2 h to facilitate chromogenic substrate conversion. Absorbance was quantified at 450 nm using an Accuris SmartReader 96 Microplate Reader (Benchmark Scientific, Sayreville, NJ, USA). Proliferation curves were plotted using GraphPad Prism software (v9.5.1; San Diego, CA, USA) based on the acquired optical density (OD) values, enabling quantitative analysis of temporal cellular proliferation dynamics.

## Plate colony formation assays

For clonogenic assays, GC cells were seeded at 500 cells/well in six-well plates (Corning, Corning, NY, USA) and cultured for 14 d (37 °C, 5% $CO_2$). Colonies were fixed with 4% paraformaldehyde (15 min), stained with 0.1% crystal violet (10 min), washed twice with PBS to remove unbound dye, and vertically air-dried before quantification.

## Flow cytometry

Apoptosis was analyzed using the Annexin V-647/Propidium iodide (PI) Dual Staining Kit (Cat# AC12L043, Life-iLab, China) following the manufacturer's optimized protocols. The experimental procedures were as follows: HGC-27 and AGS cells were gently washed with ice-cold PBS, then dissociated using Trypsin solution (0.25%) without EDTA (T1350, Solarbio, Beijing, China) at 37 °C through gradient digestion. After digestion, the cell

suspension was centrifuged at 300×g for 5 min using a Beckman Coulter Allegra X-15R centrifuge. Cell pellets were resuspended in a pre-cooled Annexin V Binding Buffer and adjusted to a density of $2 \times 10^5$ cells/mL. Each experimental group received 5 μL Annexin V-647 fluorochrome and 5 μL PI staining solution (working concentration: 50 μg/mL), with parallel preparation of unstained controls and single-color compensation controls. After gentle mixing, samples underwent light-protected incubation at 4 °C for 15 min, with strict avoidance of vortex agitation to preserve membrane integrity. The fluorescence was detected within 60 min post-staining using a CytoFLEX LX flow cytometer (Beckman Coulter, Brea, CA, USA). Apoptotic indices, defined as the cumulative percentage of early apoptotic (Annexin $V^+$ $PI^-$) and late apoptotic (Annexin $V^+$ $PI^+$) populations within the viable cell gate, were calculated using CytExpert 2.4 software with optimized doublet discrimination parameters.

### Dose-response curve

GC cells were seeded in 96-well plates at a density of $7 \times 10$ cells/well and incubated for 24 h until they reached 70% confluence. The cells were then treated with graded concentrations of 5-aza-CdR (0–40 μM) for 24–96 h. After treatment, cell viability was assessed using the CCK-8 assay (Dojindo) by adding the reagent to the culture medium at a 1:10 dilution ratio, followed by incubation at 37 °C for 1 h. Absorbance was measured at 450 nm using a microplate reader. Dose–response curves (logarithmic concentration *vs.* normalized response) were generated by fitting the data to a sigmoidal dose–response model (variable slope) using nonlinear regression analysis.

### Statistical analysis

Data processing, statistical analyses, and visualization were performed using the R software (v4.2.1). Normal distributions were analyzed using unpaired Student's t-tests, while non-normal distributions were analyzed using the Wilcoxon rank-sum test. Pearson's correlation coefficient was used to evaluate the association between two continuous variables. Given the potential influence of skewed data, Spearman's correlation analysis was conducted to ensure a comprehensive assessment of variable relationships. A two-tailed *p*-value $< 0.05$ was defined as the threshold for statistical significance.

## RESULTS

### ARID1A methylation and expression in GC

Aberrant DNA methylation patterns are a recognized epigenetic mechanism that drives oncogenic transcriptional dysregulation (*Torres et al., 2016*). To investigate the causal relationship between ARID1A aberrant expression and methylation dynamics, we performed a multi-platform bioinformatics analysis. Although ARID1A promoter methylation levels through UALCAN (*Chandrashekar et al., 2022*) showed a trend toward reduction in normal gastric mucosa, this difference did not reach statistical significance ($p > 0.05$; Fig. 1A). Genomic correlation analysis *via* cBioPortal demonstrated a robust inverse association between ARID1A methylation density and mRNA expression levels (Spearman's $\rho = -0.29$, $p = 2.06 \times 10^{-8}$; Fig. 1B). Complementary evidence from

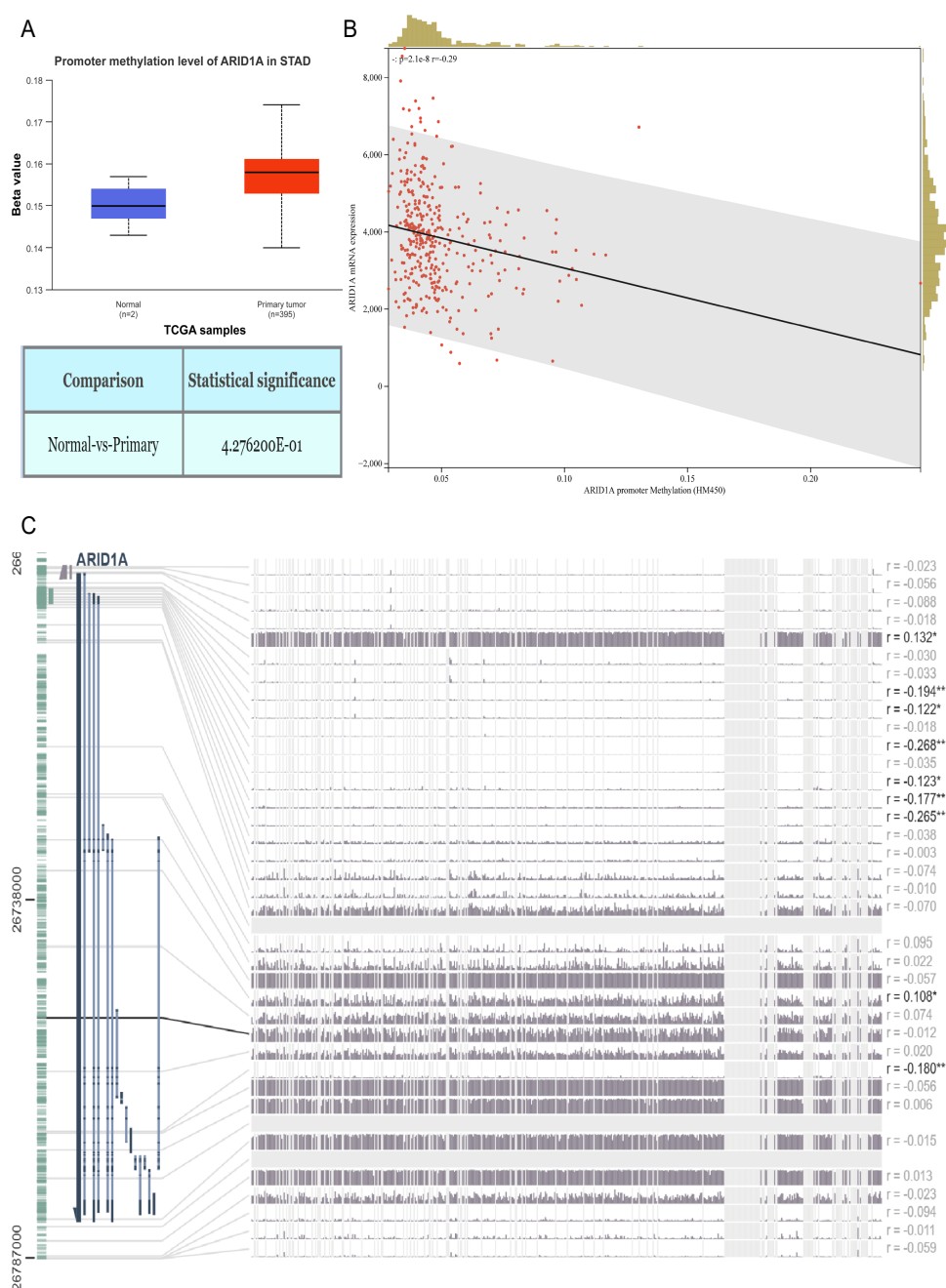

**Figure 1** **ARID1A promoter methylation is inversely associated with its expression in gastric cancer.**
(A) Comparative methylation analysis of ARID1A in gastric adenocarcinoma (STAD) *versus* normal gastric tissues using the UALCAN database. (B) Correlation between ARID1A transcriptional expression and promoter methylation in gastric cancer specimens from the cBioPortal cohort. mRNA expression levels and $\beta$-values of methylation were retrieved from the cBioPortal repository (https://github.com/cBioPortal). Linear regression analysis reveals inverse associations between methylation status and transcript abundance, with shaded bands indicating 95% confidence intervals for regression coefficients. (C) MEXPRESS

**Figure 1 (…continued)**
visualization of CpG methylation patterns within the ARID1A genomic locus correlated with gene expression in STAD. Transcript levels are depicted by blue trajectories, while adjacent heatmaps display $\beta$-values for specific CpG dinucleotides. Pearson correlation coefficients with corresponding $p$-values are annotated for significant associations (*$< 0.05$; **$< 0.01$; ***$< 0.001$).

MEXPRESS (*Koch et al., 2019*) further confirmed this anti-correlation pattern, particularly emphasizing that hypermethylation of CpG islands within the ARID1A promoter region was inversely associated with gene expression in gastric adenocarcinoma specimens (Fig. 1C). These convergent findings suggest that promoter hypermethylation serves as a potential regulatory mechanism underlying ARID1A suppression in gastric carcinogenesis.

To refine the macro-level dysregulation of methylation into site-specific microvariations, we performed additional validation using the UCSC Xena database. Stratification by ARID1A expression levels revealed significant inverse correlations between transcriptional activity and methylation at promoter-associated probes cg04081153 ($p$-value = 0.0002, FDR-adjusted $p$-value = 0.0008, Fig. 2H) and cg05445839 ($p$-value < 0.0001, FDR-adjusted $p$-value $\leq$ 0.0008, Fig. 2I). Consistent with this, samples grouped by methylation levels at these loci demonstrated lower ARID1A expression in the hypermethylated subgroups ($p$-value = 0.0001, FDR-adjusted $p$-value = 0.0001, Fig. 2J; $p$-value <0.0001, FDR-adjusted $p$-value < 0.0001, Fig. 2K). Complementary analysis *via* MEXPRESS identified 12 CpG sites exhibiting significant negative correlations with ARID1A expression (Pearson's R: −0.269 to −0.122), including cg25601319 (R = −0.194, $p$ < 0.001), cg14851949 (R = −0.122, $p$ < 0.05), cg24532126 (R = −0.268, $p$ < 0.001), cg05445839 (R = −0.123, $p$ < 0.05), cg11856093 (R = −0.177, $p$ < 0.01), and cg00371107 (R = −0.265, $p$ < 0.001) (Fig. 2A). These findings highlight cg05445839, a recurrently hypermethylated probe within the ARID1A promoter, as a potential therapeutic target for restoring physiological ARID1A expression in gastric adenocarcinoma through epigenetic modulation.

## Comprehensive profiling and functional annotation of methylation-driven gene networks in GC

Differential expression analysis between the ARID1A promoter methylation-defined subgroups (Hyper-group *vs.* Hypo-group) identified 191 methylation-regulated differentially expressed genes (MeDEGs), comprising 37 upregulated and 154 downregulated genes meeting the established significance thresholds (FDR-adjusted $p$-value < 0.05, |$\log_2$FC| $\geq$ 1.5) (Table S1). Transcriptional alterations were visualized using a volcano plot highlighting the MeDEG distribution (Fig. 3A), while hierarchical clustering of the top 50 statistically significant MeDEGs demonstrated robust subgroup segregation (Fig. 3B). Furthermore, a correlation matrix delineated the co-expression patterns between the 10 most significantly upregulated and downregulated MeDEGs, revealing potential regulatory networks influenced by ARID1A promoter hypermethylation (Fig. S1A). To elucidate the biological significance of the MeDEGs in gastric carcinogenesis, we performed integrative GO and KEGG pathway enrichment analyses on the 191 identified candidates. GO analysis revealed nine enriched BP, eight CC, and nine MF, with cell adhesion, apoptotic regulation, and signaling receptor

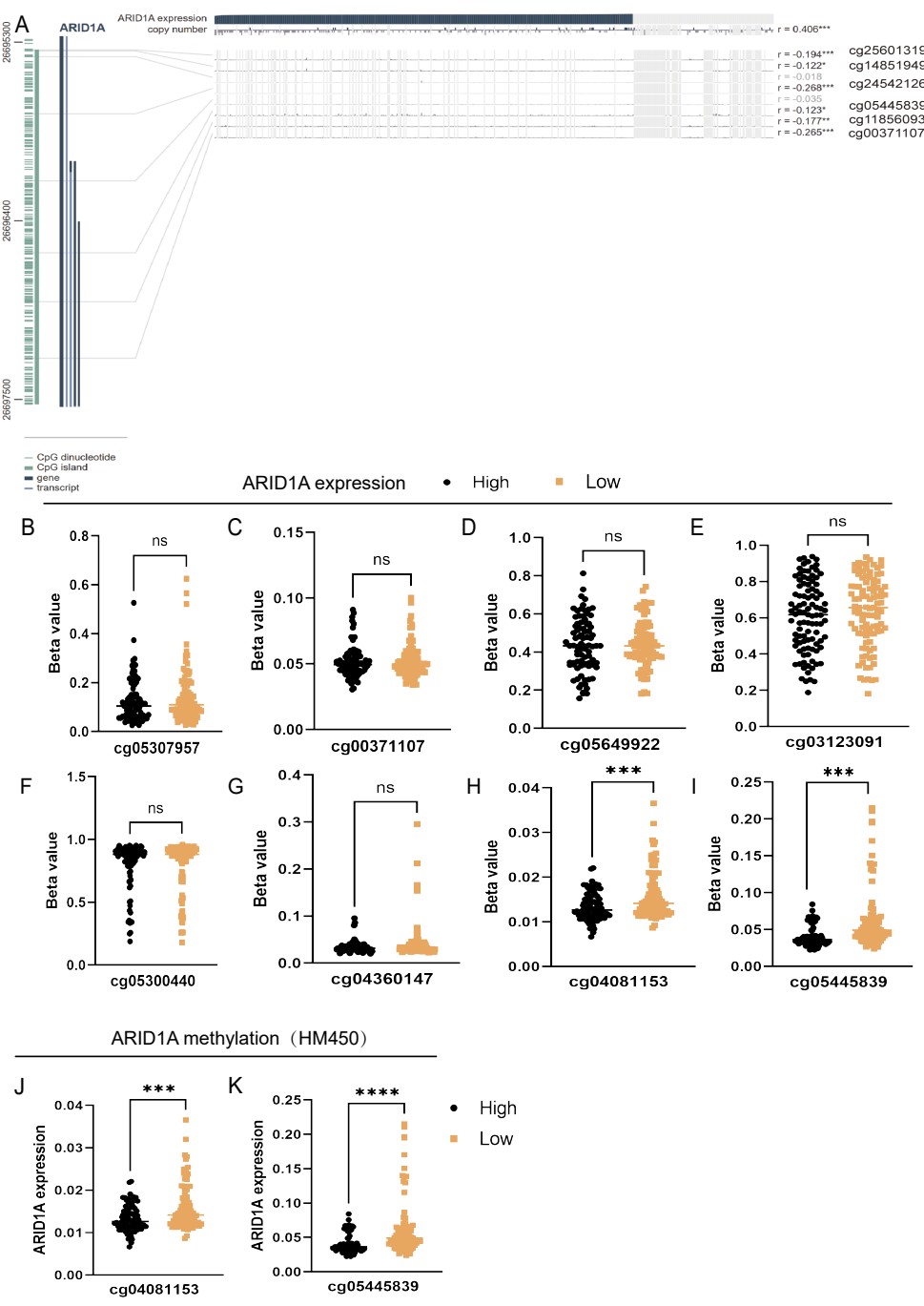

**Figure 2 Site-specific methylation profiling of ARID1A identifies hypermethylated CpG sites in gastric cancer.** (A) Promoter-specific analysis of the ARID1A locus. A subset of the genomic region shown in Fig. 1C, focusing on CpG islands within the promoter. Significantly hypermethylated probes are explicitly labeled with their probe names. (B–I) Promoter-associated methylation profiling of ARID1A in the UCSC Xena cohort. Quartile-based stratification of ARID1A expression revealed significant inverse correlations between transcriptional activity and methylation $\beta$-values at probescg04081153 and cg0544839, whereas 

**Figure 2 (…continued)**
methylation levels at other interrogated loci showed no association with gene expression. (J–K) Parallel analysis stratifying samples by methylation status atcg04081153 and cg0544839 demonstrated reduced ARID1A expression in hypermethylated subgroups. (ns, not significant; ***< 0.001; ****< 0.0001).

activity identified as the most significantly enriched terms, along with immune-related pathways (Fig. 3C). KEGG pathway mapping further demonstrated the predominant enrichment of oncogenic signaling cascades, including cell-matrix adhesion, PI3K-AKT activation, and mTOR-mediated proliferation (Fig. 3D). PPI network reconstruction using STRING and Cytoscape identified key regulatory hubs (Fig. S1B), with the top 10 centrality-ranked nodes (closeness metrics) highlighting potential master regulators (Fig. 3E). Modular decomposition using the MCODE algorithm resolved two functionally cohesive subnetworks implicated in distinct carcinogenic processes (Figs. 3F–3G). Comparative analysis of simple nucleotide variation data from TCGA revealed distinct mutational architectures between ARID1A hypermethylated and hypomethylated GC cohorts. In the hypermethylated subgroup, the five most frequently mutated genes were *TTN* (54.9%), *TP53* (45.1%), *MUC16* (34.1%), *LRP1B* (26.8%), and *ARID1A* (25.6%) (Fig. S1C). Conversely, the hypomethylated cohort exhibited predominant alterations in *TTN* (53.3%), *TP53* (50.7%), *LRP1B* (32.0%), *MUC16* (30.7%), and *CSMD3* (28.0%) (Fig. S1D).

## GSEA of MeDEGs

Stratification based on ARID1A promoter methylation $\beta$-values identified 16,030 MeDEGs. GSEA revealed significant associations with hallmark oncogenic pathways, including apoptotic regulation, immune response, NF-kappa B signaling, and T-cell receptor signaling, with NES and $p$-value annotated for each pathway (Figs. 4A–4L).

## ARID1A methylation and tumor immune microenvironment modulation

Integrated analysis of the TISIDB database revealed significant correlations between ARID1A expression and chemokine/chemokine receptor dynamics in GC (Figs. 5A–5B). We found that ARID1A expression positively correlated with several chemokine/chemokine receptor genes, as illustrated in Figs. 5C–5G: CXCL14 ($r = 0.102$, $p = 0.0374$), CCR7 ($r = 0.134$, $p = 0.00616$), CCR9 ($r = 0.105$, $p = 0.032$), CXCR5 ($r = 0.131$, $p = 0.0075$), and XCR1 ($r = 0.103$, $p = 0.0363$). Conversely, ARID1A levels were inversely associated with a broad spectrum of chemokine ligands and their receptors (Figs. S2A–S2T).

ARID1A expression demonstrated a complex immunoregulatory signature in GC, as overviewed in the correlation heatmap (Figs. 5H–5I). Specifically, ARID1A levels showed significant positive correlations with several key immunomodulatory molecules, including the immunostimulators CD40LG ($r = 0.102$, $p = 0.037$; Fig. 5J), TNFRSF13B ($r = 0.11$, $p = 0.0249$; Fig. 5K), TNFRSF13C ($r = 0.244$, $p = 5.11e-7$; Fig. 5L) and TNFRSF25 ($r = 0.18$, $p = 0.000228$; Fig. 5M), as well as the immunoinhibitors ADORA2A ($r = 0.146$, $p = 0.00294$; Fig. 5N) and CD160 ($r = 0.154$, $p = 0.00168$; Fig. 5O). Conversely, ARID1A was inversely associated with a broader set of immunomodulatory molecules (*e.g.*,

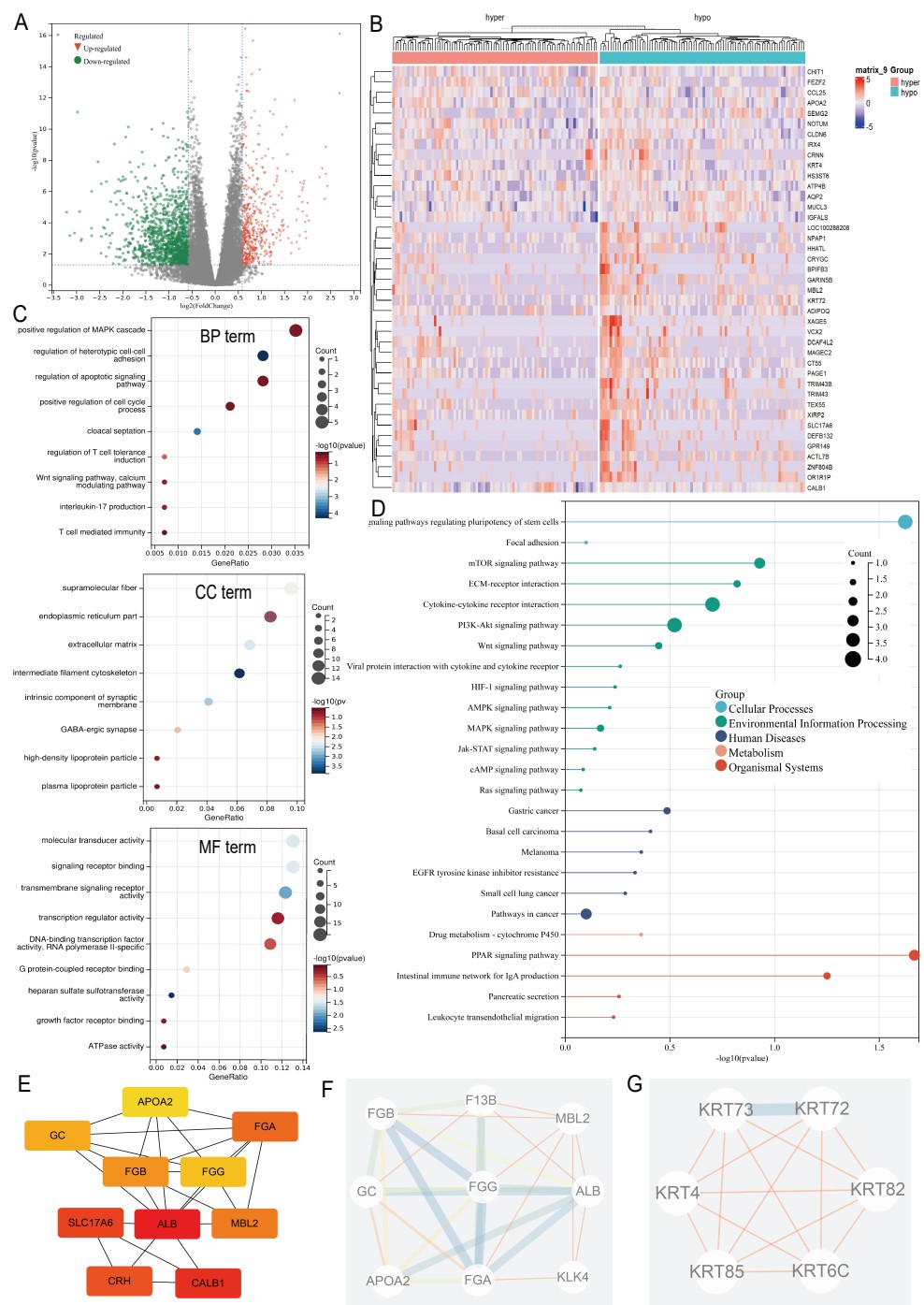

**Figure 3** **Multi-omic profiling of methylation-regulated gene networks in gastric cancer.** (A) Volcano plot of MeDEGs between hyper- and hypomethylated subgroups (|log2 fold change (FC)| > 1.5, $P < 0.05$). Red/green/black nodes denote upregulated, downregulated, and non-significant genes, respectively. (B) Hierarchically clustered heatmap of the top 50 MeDEGs (|log2FC| > 1.5, $P < 0.05$), with dendrograms indicating sample clustering. (C) Gene Ontology (GO) enrichment analysis of MeDEGs, highlighting significant terms in biological processes (BP), cellular components (CC), and molecular functions

**Figure 3 (...continued)**
(MF). Node size corresponds to the number of enriched genes, while color intensity reflects the statistical significance (*P*-value). (D) Kyoto Encyclopedia of Genes and Genomes (KEGG) pathway enrichment. Key cancer-related pathways are annotated. (E) Top 10 hub genes identified by closeness centrality analysis. (F–G) Two functionally distinct modules extracted *via* MCODE algorithm, representing cohesive molecular subnetworks.

HAVCR2, IL-10, LGALS9, CD276; complete details of all significant negative correlations are provided in Figs. S3A–S3N). This bidirectional immunoregulatory signature establishes ARID1A as a key modulator of tumor-immune interactions in gastric cancer.

To investigate the influence of ARID1A promoter hypermethylation on immunomodulation, we systematically investigated immune infiltration patterns across methylation-defined subgroups using seven computational algorithms. CIBERSORT deconvolution revealed notable differences in lymphoid populations; the hypomethylated subgroup exhibited elevated infiltration of resting memory CD4$^+$ T cells ($p < 0.0001$) and resting dendritic cells ($p < 0.05$), along with reduced M0 macrophage abundance ($p < 0.0001$). Consensus across the MCP-counter, TIMER, and Quantiseq algorithms identified significant B-cell infiltration divergence between subgroups, suggesting methylation-dependent regulation of humoral immunity. Stratification using the ESTIMATE algorithm revealed higher immune scores ($p < 0.05$) and ESTIMATE scores ($p < 0.05$), along with lower tumor purity ($p < 0.05$) in the hypermethylated cohort. Immunophenoscore (IPS) analysis further revealed subgroup-specific disparities in CP and AZ compartments (Fig. 6A).

Given the significant divergence in immune infiltration between the two clusters, we further evaluated their association with canonical immune checkpoints. Immune checkpoint analysis revealed distinct expression patterns of CTLA-4 ($p < 0.05$) and BTLA ($p < 0.05$) across methylation-defined subgroups (Fig. 6B). Emerging evidence has highlighted the interaction between tumor-associated sialoglycans and sialic acid-binding immunoglobulin-like lectins (Siglecs) in infiltrating immune cells as a novel immune checkpoint axis, offering therapeutic potential in cancer immunotherapy (*Stanczak & Läubli, 2023*). Supporting this paradigm, *Haas et al. (2019)* demonstrated that Siglec-7 and Siglec-9 suppress T cell activation upon TCR(T-cell receptor) stimulation (*Stanczak et al., 2018*). Notably, our Siglec profiling identified elevated expression of Siglec-7 ($p < 0.01$) and Siglec-9 ($p < 0.001$) in the hypermethylated subgroup compared to their hypomethylated counterparts, suggesting methylation-dependent regulation of this immunosuppressive axis (Fig. 6B).

## Promoter hypermethylation inhibits ARID1A expression in gastric cancer cells

Epigenetic modifications constitute one of the primary mechanisms underlying gene silencing (*Long et al., 2023*). To determine whether ARID1A can be regulated by DNA methylation, we examined the ARID1A promoter sequence using the NCBI database and predicted it with the MethPrimer database. Bioinformatics prediction using the MethPrimer identified three hypermethylated CpG islands in the ARID1A promoter

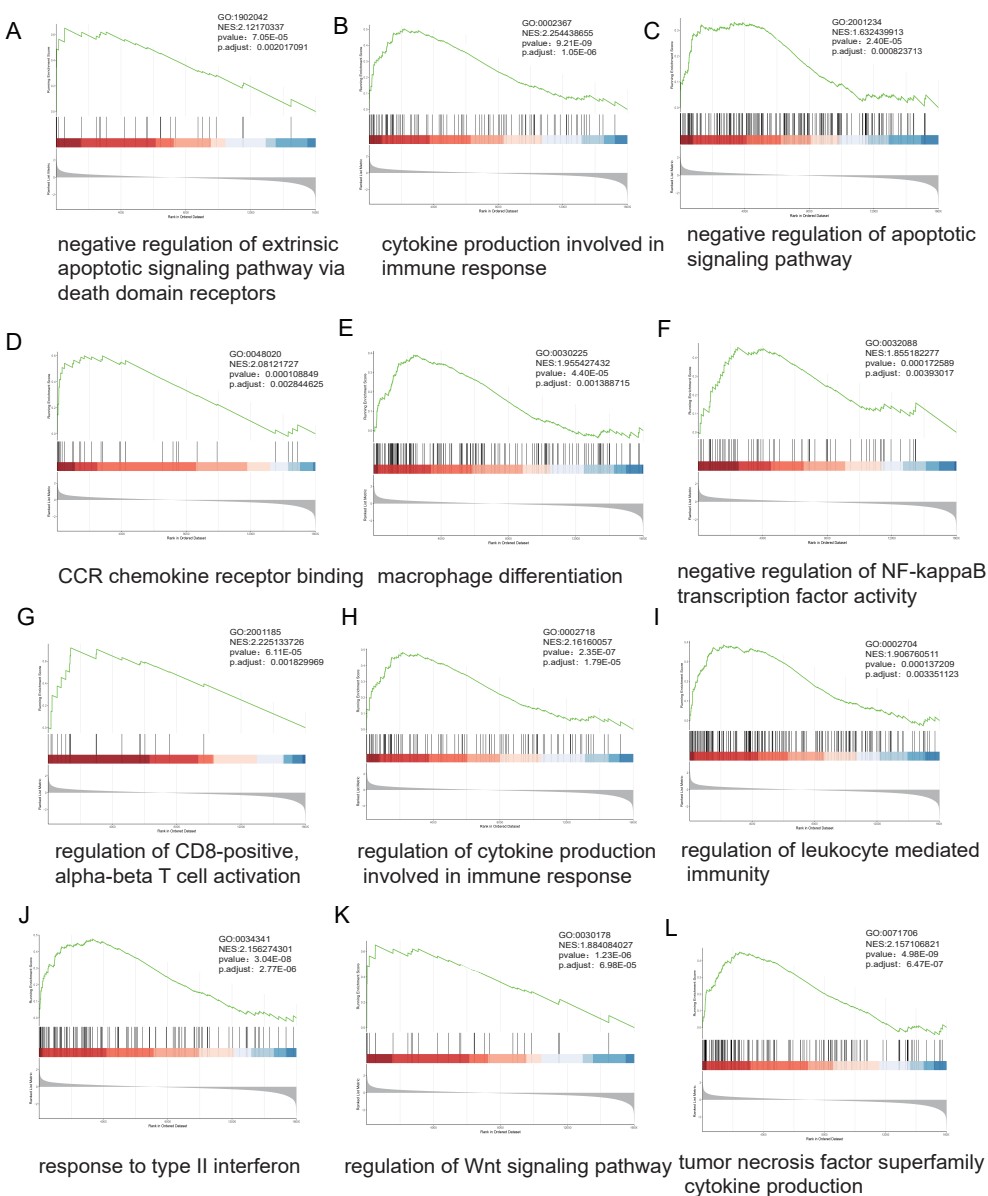

**Figure 4 Genome-wide enrichment analysis of methylation-regulated pathways.** (A–L) Gene Set Enrichment Analysis (GSEA) profiles depicting methylation-driven pathway alterations. Each panel comprises: Enrichment score (ES) trajectory: Peaks represent maximal ES values for hallmark gene sets. Gene ranking distribution: Black vertical lines indicate positions of MeDEGs in the ranked gene list; leading-edge subsets (green highlight) denote genes contributing most significantly to enrichment. Rank metric distribution: Gray-scale heatmaps display ranked gene expression differences (red: hypermethylated subgroup overexpression; blue: hypomethylated subgroup overexpression), with corresponding signal-to-noise ratios (SNRs) annotated.

region (Fig. 7A). Among these, we selected a CpG island adjacent to the transcription start site (TSS: chr1:26692500) for primer design and subsequent validation using MSP. MSP detected exclusive amplification of methylated ARID1A promoter fragments in HGC-27

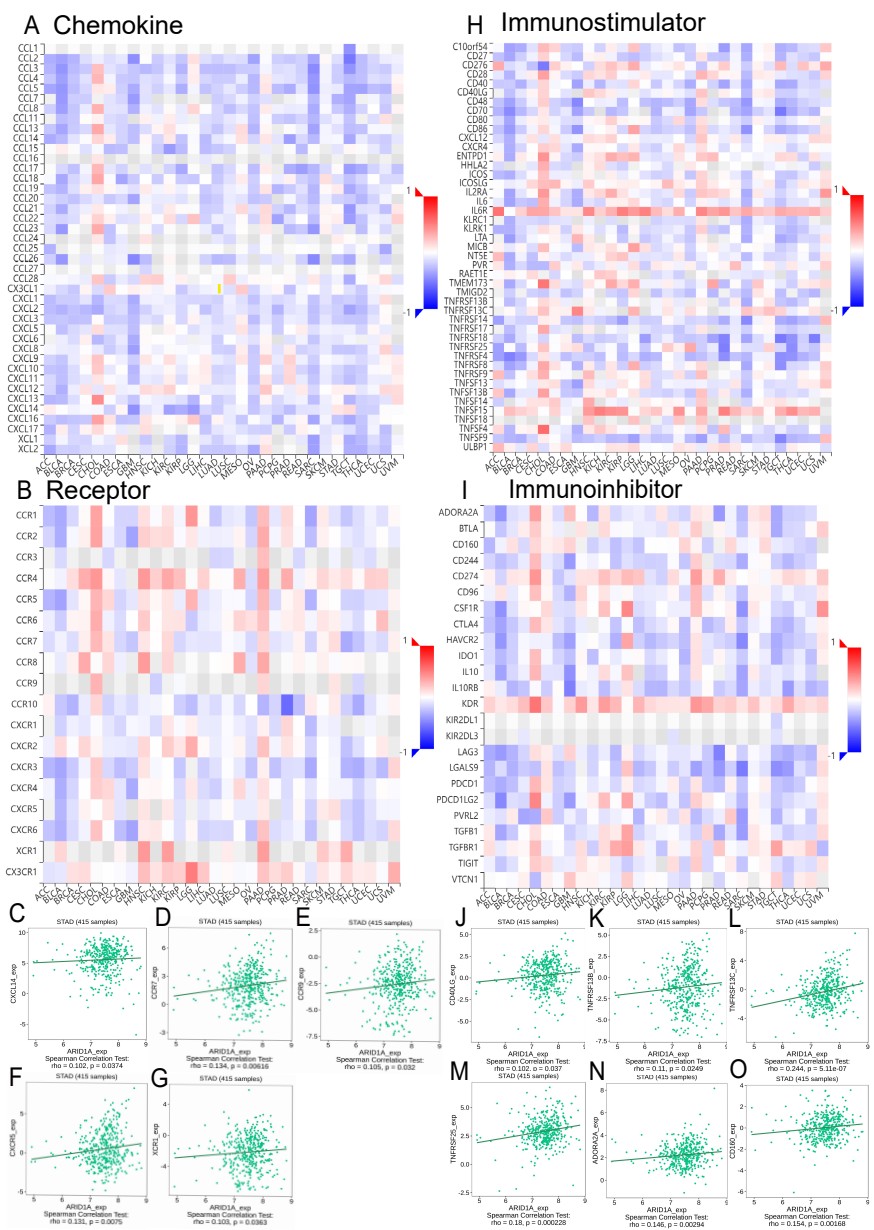

**Figure 5 Correlation analysis of ARID1A with chemokine networks and immunomodulators using the TISIDB database.** (A) Heatmap analysis of correlations between ARID1A expression and chemokine ligands in gastric cancers. (B) Heatmap analysis of correlations between ARID1A expression and chemokine receptors in gastric cancers. (C–G) Positive correlations (Spearman) between ARID1A expression and specific chemokine/chemokine receptors. (H) Heatmap visualization of ARID1A correlations with immunostimulator in the tumor microenvironment. (D) Heatmap visualization of ARID1A correlations with immunoinhibitor in the tumor microenvironment. (J–O) Significant positive correlations (Spearman) between ARID1A expression and immunomodulatory molecules in gastric cancer.

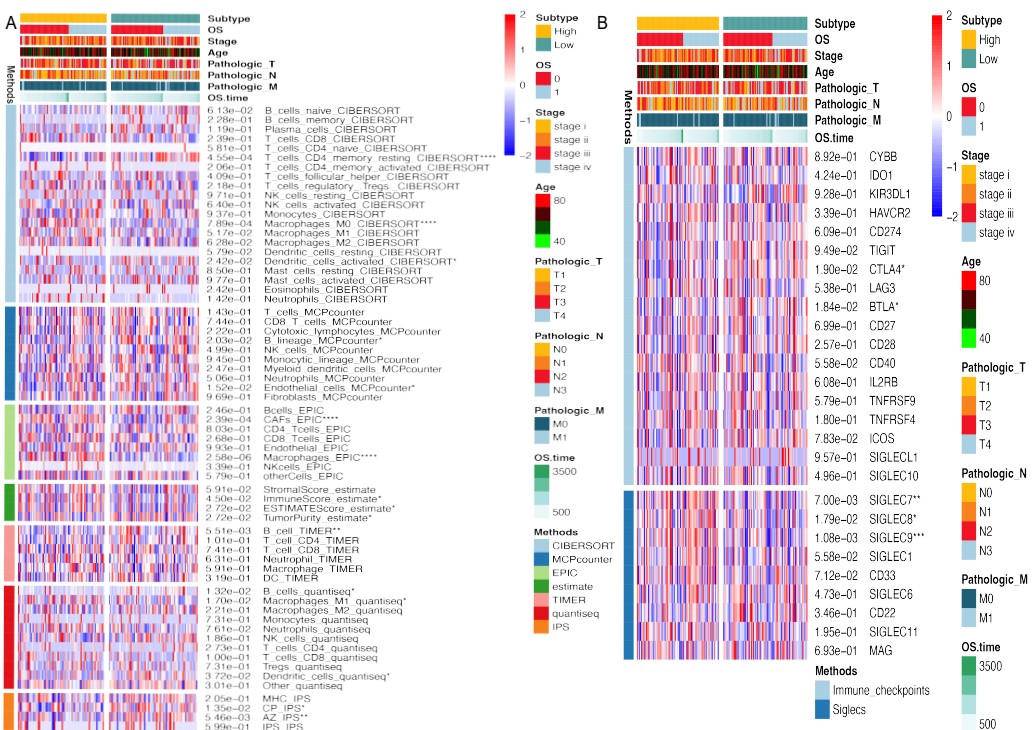

**Figure 6 Immune microenvironment profiling stratifed by ARID1A methylation status in gastric cancer.** (A) Multi-algorithmic profiling of immune microenvironment in ARID1A methylation-defined GC Subgroups. (B) Methylation-defined immune checkpoint and Siglec expression profiles in GC subgroups.

cells, while AGS cells exhibited both methylated and unmethylated amplification products (Fig. 7B). Dose–response profiling across a 0–40 μM 5-aza-CdR gradient demonstrated concentration-dependent suppression of HGC-27 proliferation (IC50 = 18.57 μM; Fig. 7E). Conversely, AGS cells showed no significant proliferation suppression even at the maximum tested concentration (40 μM) of 5-aza-CdR (Fig. S4A). Given the biallelic methylation pattern (partial promoter methylation) (Fig. 7B) and low baseline ARID1A expression in AGS cells (Figs. 7C–7D), pharmacological demethylation with 5-aza-CdR failed to significantly restore ARID1A expression (Fig. S4B). This resistance likely stems from incomplete epigenetic silencing, as evidenced by residual unmethylated alleles that permit constitutive low-level transcription. Conversely, HGC-27 cells exhibited complete ARID1A promoter hypermethylation coupled with reversible transcriptional silencing (post-5-aza-CdR), making them an optimal model for investigating methylation-dependent oncogenic mechanisms.

CCK-8 proliferation assays revealed significant suppression of HGC-27 cell proliferation at 10 μM 5-aza-CdR ($p < 0.01$, Fig. 7F). This observation suggests that maximal epigenetic efficacy is achieved at 10 μM, beyond which cytotoxicity-driven effects dominate. Consequently, 10 μM was chosen for subsequent experiments to avoid nonspecific toxicity while maintaining effective target modulation. Time- and concentration-resolved analyses identified 10 μM 5-aza-CdR with 48 h exposure as the optimal regimen, achieving maximal

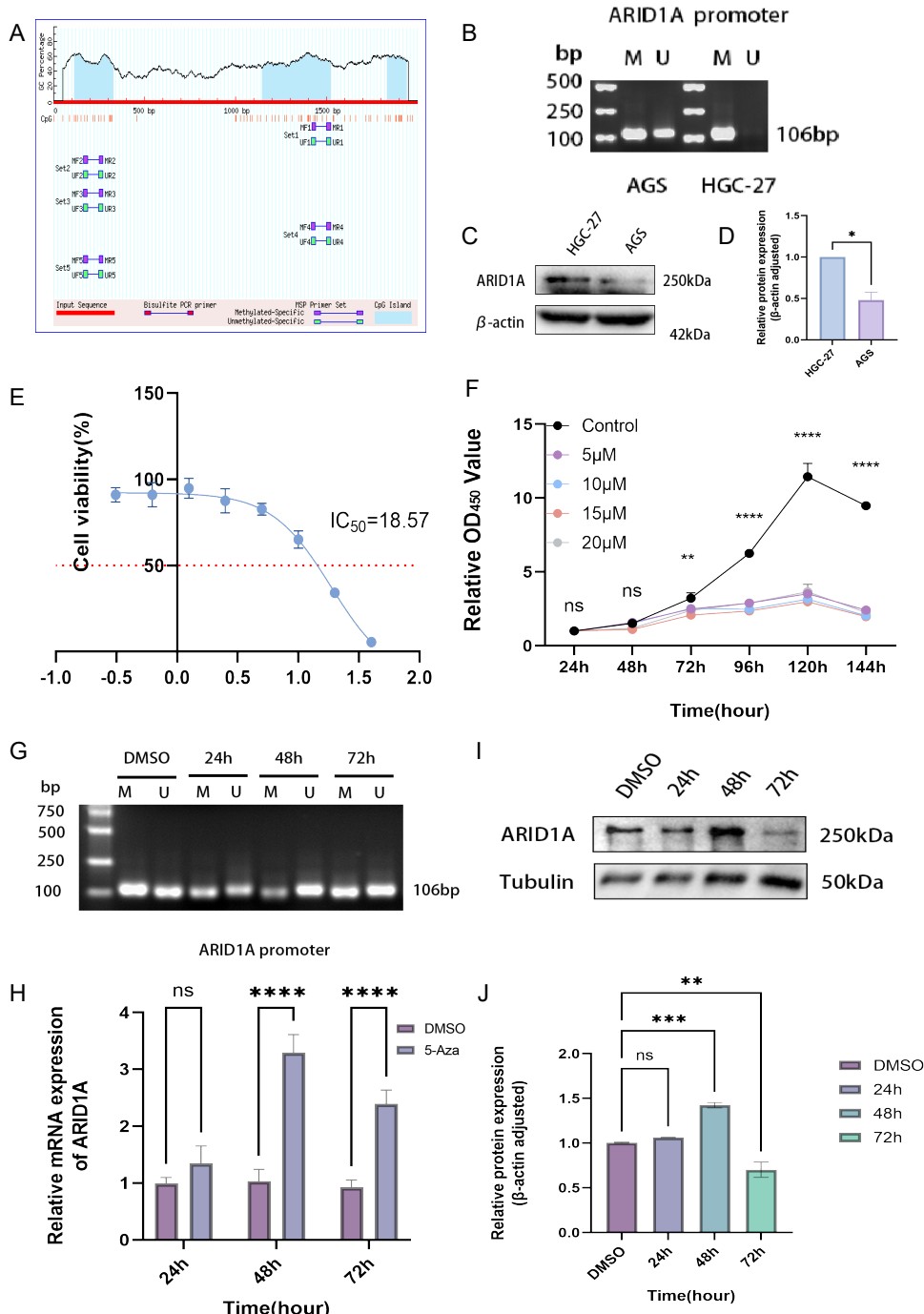

**Figure 7** **Epigenetic modulation and therapeutic targeting of ARID1A in gastric cancer.** (A) Predicted CpG islands in the ARID1A promoter region. The vertical coordinate is the percentage of GC content in the promoter region, the horizontal coordinate is the length of the promoter sequence, and the blue area represents the CpG island. (B) MSP was used to detect the methylation status of the ARID1A promoter in

**Figure 7 (…continued)**
GC cells, with M indicating methylation and U indicating unmethylation. The cell experiment was performed three times. MSP, methylation-specific PCR. (C–D) Comparison of ARID1A expression in AGS and HGC-27 cells *via* western blot (protein level). (E) Dose–response curve of 5-aza-CdR in HGC-27 (IC50 = 18.57 μM). (F) The CCK-8 assay revealed that 5-aza-CdR (5–20 μM) inhibited the proliferation of HGC-27 cells in a dose-dependent manner. Significant inhibition occurred at 10 μM, but higher doses provided little additional suppression. (G) Methylation-specific PCR (MSP) analysis of ARID1A promoter methylation in HGC-27 cells treated with 10 μM 5-aza-CdR for 24, 48, or 72 h. Agarose gel electrophoresis shows a time-dependent reduction of methylated (M) alleles and a corresponding increase in unmethylated (U) alleles, reaching maximal demethylation at 48 h. Notably, a rebound in methylation levels is observed after 72 h of prolonged treatment. (H) Quantitative PCR(qPCR) analysis of ARID1A expression in HGC-27 cells treated with 10 μM 5-aza-CdR for 24, 48, or 72 h. Maximal transcriptional induction was observed at 48 h. (I–J) Western blot and quantification confirms transient ARID1A protein upregulation at 48 h, diminishing thereafter. Tublin was used as a loading control. Statistical significance is indicated as follows: ns, not significant ($p \geq 0.05$); *$p < 0.05$; **$p < 0.01$; ***$p < 0.001$; ****$p < 0.0001$.

ARID1A transcriptional reactivation (Fig. 7H) and promoter demethylation (Fig. 7G). Prolonged treatment (>48 h) paradoxically increased methylation (Fig. 7G) and reduced ARID1A protein levels (Figs. 7I–7J), indicating a potential escape from the initial epigenetic modulation.

## ARID1A depletion promotes GC progression

To evaluate the tumor-suppressive role of ARID1A in GC, we performed stable ARID1A knockdown in HGC-27 and AGS cell lines. Western blot and qPCR analyses confirmed significant reductions in ARID1A protein (Figs. 8A–8B) and mRNA levels (Fig. 8C) in the HGC-27 cell line, with similar trends observed in AGS cells (Figs. S5A–S5C). Transwell assays demonstrated that ARID1A silencing enhanced the migratory ($p < 0.0001$) and invasive capacities significantly ($p < 0.0001$) in HGC-27 cells (Figs. 8D–8F). Consistent findings were observed in AGS cells with a significant increase in migration ($p < 0.0001$) and invasion ($p < 0.001$) post-knockdown (Figs. S5D–S5F). Wound healing assays further corroborated the accelerated migration rates in both ARID1A-depleted cell lines (HGC-27: $p < 0.05$; AGS: $p < 0.0001$) (Figs. 8G–8H, Figs. S5G–S5H). CCK-8 proliferation assays revealed significant growth potentiation in ARID1A-deficient GC cells (Fig. 8L; Fig. S5L). Long-term clonogenic survival assays confirmed a sustained proliferative advantage (HGC-27: $p < 0.01$; AGS: $p < 0.0001$) (Figs. 8I–8J, Figs. S5I–S5J). Flow cytometric analysis using Annexin V-647/PI staining showed significant suppression of apoptosis in ARID1A-silenced cells (HGC-27: $p < 0.01$; AGS: $p < 0.01$) (Figs. 8K, 8M; Figs. S5K, S5M). These findings mechanistically link ARID1A loss to proliferative enhancement and apoptotic resistance in GC progression.

## Functional impact of ARID1A demethylation in gastric cancer

Pharmacological demethylation with 10 μM 5-aza-CdR in HGC-27 cells reversed ARID1A promoter hypermethylation, significantly impairing metastatic potential. Transwell assays indicated a marked reduction in both migration and invasion compared to controls (Figs. 9A–9C). Clonogenic survival assays showed greatly reduced colony formation (Figs. 9D–9E). This anti-proliferative effect was consistent with the substantial growth suppression revealed by CCK-8 assays (Fig. 9H). Flow cytometry analysis demonstrated

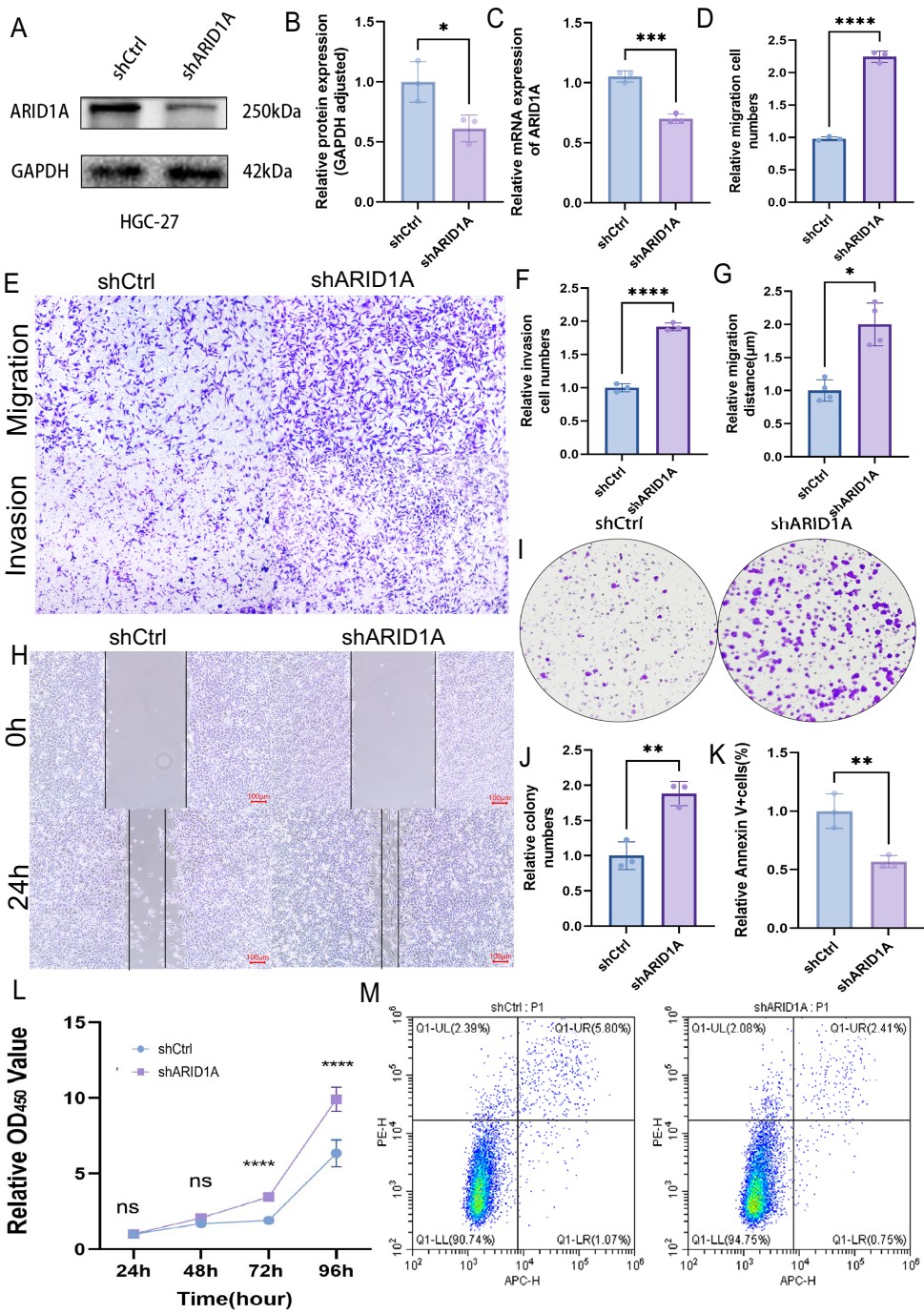

**Figure 8** **ARID1A knockdown promotes malignant phenotypes in gastric cancer HGC-27 cells.** (A–C) Validation of ARID1A knockdown efficiency in HGC-27 cells *via* Western blot (protein level) and qPCR (transcript level). (D–F) Transwell invasion and migration assays (10x magnification) demonstrating enhanced metastatic potential in ARID1A-deficient HGC-27 cells, visualized by crystal violet staining and analyzed using GraphPad Prism 9.0. (G–H) Wound-healing assays quantifying accelerated migration rates in ARID1A-silenced HGC-27 cells. 

**Figure 8 (…continued)**
(I–J) Colony formation assays confirming enhanced clonogenic survival in ARID1A-depleted HGC-27 cells. (L) CCK-8 proliferation assays showing increased growth kinetics post-ARID1A knockdown. (K, M) Flow cytometry analysis of apoptosis resistance in ARID1A-silenced HGC-27 cells. Quantification dataanalyzed using GraphPad Prism 9.0 (*$p < 0.05$, **$p < 0.01$, ***$p < 0.001$, ****$p < 0.0001$).

a significant increase in apoptosis relative to control cells (Figs. 9F–9G). Together, these results identify ARID1A promoter hypermethylation as a critical driver of gastric cancer progression and highlight the therapeutic potential of 5-aza-CdR in HGC-27 cells *via* promoting apoptosis and suppressing proliferation and invasion.

### ARID1A methylation drives immune evasion *via* PD-L1 upregulation and PI3K/AKT/mTOR activation in GC

PD-L1, a key immunosuppressive checkpoint molecule, is elevated in GC and is correlated with adverse clinical outcomes (*Wu et al., 2006*; *Hou et al., 2014*). Previous studies have demonstrated elevated PD-L1 expression in ARID1A-mutated tumors (*Kim et al., 2019*). In our experimental models, ARID1A knockdown in HGC-27 cells induced an increase in PD-L1 expression ($p < 0.01$) (Figs. 10A–10C), with similar PD-L1 upregulation observed in AGS cells ($p < 0.0001$) (Figs. S6A–S6C). Notably, pharmacological demethylation of HGC-27 cells with 5-aza-CdR restored ARID1A expression while suppressing PD-L1 (Figs. 10D–10F).These findings suggest an inverse regulatory relationship between ARID1A and PD-L1 in the studied models, implying that ARID1A loss-of-function may reprogram immune checkpoint signaling in gastric cancer.

Based on our earlier profiling data that identified PI3K–AKT–mTOR signaling as enriched in ARID1A-hypermethylated tumors (Fig. 3D), we further investigated its role as a mechanistic link. Transcriptomic analysis of TCGA cohorts confirmed PI3K/AKT pathway activation in ARID1A-hypermethylated tumors, marked by elevated expression of PIK3R1, PIK3R2, PIK3CA, and AKT3 compared to hypomethylated controls (Figs. S6E, S6F, S6G, and S6J, respectively). Accordingly, the restoration of ARID1A expression by 5-aza-CdR treatment in HGC-27 cells suppressed PI3K/AKT/mTOR pathway activation, as evidenced by reduced phosphorylation of PI3K, AKT, and mTOR (Figs. 10D, 10G–10L).

To functionally validate this mechanism, we performed a rescue assay in which SC79, an AKT agonist, was applied to 5-aza-CdR-demethylated HGC-27 cells, resulting in reactivated AKT/mTOR phosphorylation and restored PD-L1 expression (Figs. 10M–10S). A comprehensive rescue experiment involving four experimental groups further established the functional hierarchy of the ARID1A–PI3K/AKT–PD-L1 axis (Figs. 10T–10V): the control group showed baseline ARID1A and PD-L1 levels, SC79 alone enhanced PD-L1 expression confirming pathway sufficiency, 5-aza-CdR alone restored ARID1A and suppressed PD-L1, and most crucially, the addition of SC79 reversed 5-aza-CdR-induced PD-L1 suppression, confirming that PI3K/AKT activation is necessary for PD-L1 upregulation following ARID1A methylation.

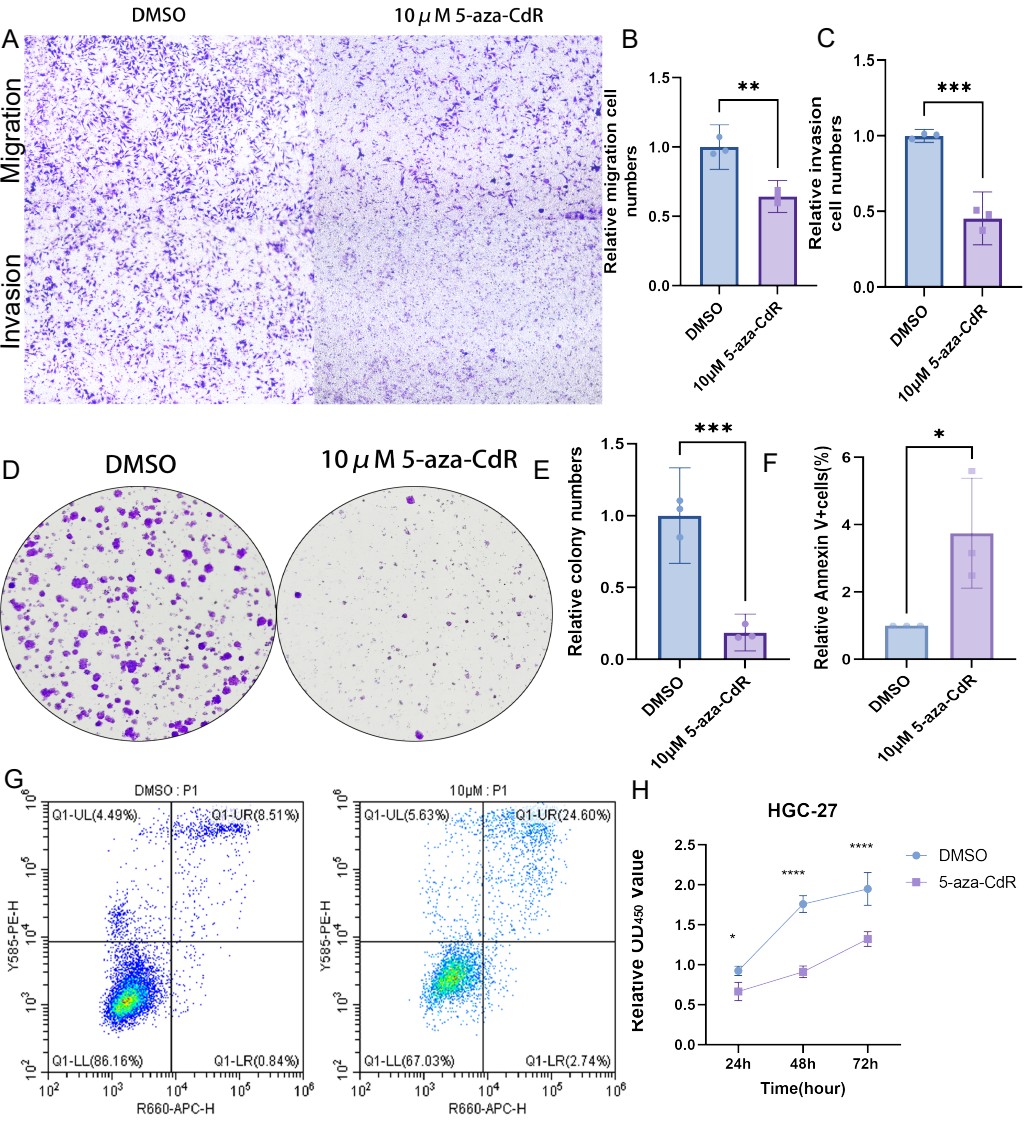

**Figure 9** **Functional impact of ARID1A demethylation in gastric cancer.** (A–C) Transwell assays demonstrate reduced migration and invasion capabilities in 5-aza-CdR-treated HGC-27 cells, with quantitative analysis performed using GraphPad Prism. (D) Clonogenic survival assays confirm suppression of colony formation in 5-aza-CdR-treated HGC-27 cells ($p < 0.001$). (F–G) Flow cytometry reveals elevated apoptosis in 5-aza-CdR-treated HGC-27 cells ($p < 0.05$). (H) CCK-8 assays demonstrate proliferation inhibition after 5-aza-CdR. Statistical significance is indicated as follows: ns, not significant ($p \geq 0.05$); *$p < 0.05$; **$p < 0.01$; ***$p < 0.001$; ****$p < 0.0001$.

# DISCUSSION

Emerging evidence has implicated ARID1A dysregulation in tumor progression through multifaceted interference with DNA damage repair mechanisms (*Mullen et al., 2021*). ARID1A is frequently mutated in GC, with mutation frequencies ranging from 14–24% (*Wang et al., 2011*). These alterations are primarily characterized by nonsense and

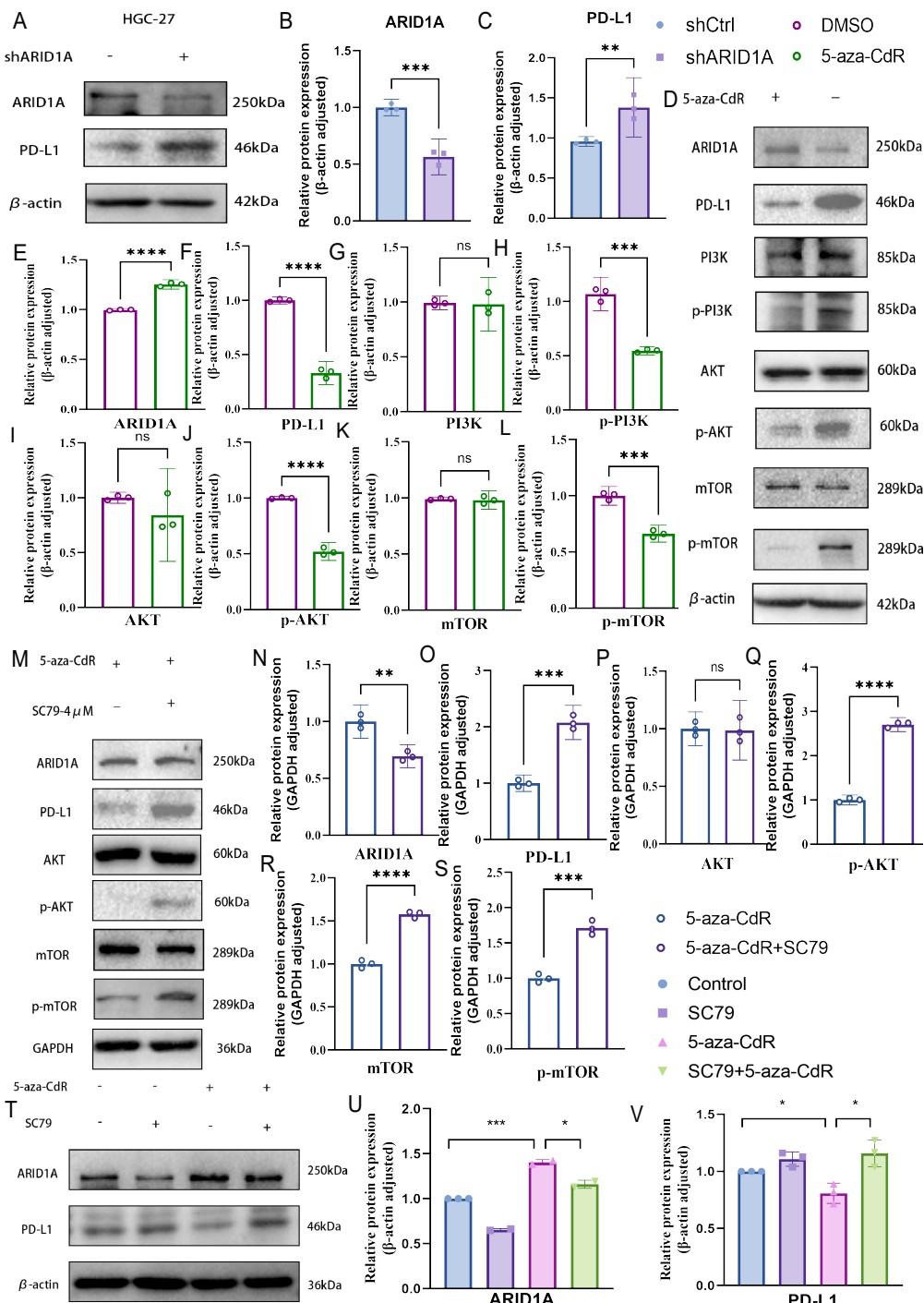

**Figure 10  ARID1A hypermethylation activates PI3K/AKT/mTOR signaling in gastric cancer.** (A–C) Western blot analysis of ARID1A and PD-L1 expressions inHGC-27 cells. (D–L) Western blot analysis of ARID1A, PD-L1 and PI3K/AKT/mTOR pathway proteins in HGC-27 cells treated with 5-aza-CdR. (M–S) Western blot analysis of ARID1A, PD-L1, and AKT/mTOR pathway proteins in HGC-27 cells treated (continued on next page...)

**Figure 10 (…continued)**
with 5-aza-CdR in the presence or absence of the AKT agonist SC79. (T–V) Western blot analysis of ARID1A and PD-L1 expression across the four experimental groups (untreated control, SC79 only, 5-aza-CdR only, and 5-aza-CdR + SC79 combination) demonstrates the functional relationship between ARID1A loss, PI3K/AKT signaling, and PD-L1 upregulation. Statistical significance is indicated as follows: ns, not significant ($p \geq 0.05$); *$p < 0.05$; **$p < 0.01$; ***$p < 0.001$; ****$p < 0.0001$.

frameshift mutations that disrupt the ARID1A protein function or expression. Notably, most studies have attributed ARID1A protein loss to genetic alterations (*Angelico et al., 2024*), leaving its epigenetic regulation largely unexplored in gastric carcinogenesis. Our study reveals promoter hypermethylation as a novel and distinct mechanism of ARID1A inactivation in GC. This epigenetic silencing drives STAD progression independently of mutational events, highlighting a previously unrecognized dimension of ARID1A dysregulation. These findings challenge the conventional genome-centric paradigm and provide critical insights into the interplay between epigenetic and genetic drivers in gastrointestinal malignancies.

Aberrant DNA methylation, a hallmark epigenetic alteration in malignancies, plays a key role in oncogenic progression (*Wu et al., 2023*). This study establishes ARID1A promoter hypermethylation as a key epigenetic silencing mechanism in gastric carcinogenesis. cBioPortal analyses revealed significant ARID1A transcriptional downregulation in GC tissues, strongly correlated with promoter CpG island hypermethylation (Spearman's $\rho = -0.29$, $p = 2.06 \times 10^{-8}$). While UALCAN analysis of TCGA data showed no significant methylation difference at the whole-promoter level (Fig. 1A), this likely reflects: 1. limited statistical power to detect subtle effects in public datasets; 2. Technical variations across source datasets may mask true biological differences. Bioinformatic prediction identified significant differentially methylated CpG sites within the ARID1A promoter region (chr1:26691236–26693235, TSS:26692500). MSP confirmed elevated methylation frequencies in HGC-27 and AGS cell lines, while pharmacological demethylation using 5-aza-CdR reduced methylation levels and restored ARID1A protein expression, establishing direct epigenetic causality.

Notably, 5-aza-CdR treatment induced significant phenotypic reprogramming in GC cells, including proliferation arrest, impaired invasiveness, and apoptosis reactivation. This effect was mediated through promoter demethylation of *ARID1A*, restoring its tumor-suppressive function to inhibit oncogenic transformation and enhance antigen presentation. Concurrently, as shown by *Li et al. (2023)*, low-dose 5-aza-CdR synergizes with PD-1 blockade by hypomethylating exhaustion-related genes (*e.g., JunD*), thereby sustaining the clonal expansion of tumor-reactive CD8$^{+}$ T cells and preserving their progenitor-like proliferative capacity. Therefore, combining 5-aza-CdR with PD-1 inhibitors presents a promising therapeutic strategy for patients with GC exhibiting *ARID1A* promoter methylation.

To elucidate the mechanistic link between ARID1A hypermethylation and gastric carcinogenesis, we employed an integrative multi-omics strategy that combined bioinformatics prediction and functional validation. KEGG pathway enrichment analysis revealed significant activation of the "PI3K-AKT and mTOR signaling pathways" in

ARID1A-hypermethylated tumors, suggesting their involvement in ARID1A-mediated oncogenesis. The PI3K/AKT/mTOR axis is hyperactivated in advanced gastric cancer (*Zhong et al., 2024*). This drives tumor progression by enhancing proliferation (*Li et al., 2021*), metastasis (*Wang et al., 2021*), and critically establishing immune evasion (*Koh et al., 2021*). While PI3K/AKT/mTOR activation typically stems from genetic lesions (*e.g.*, ARID1A mutations) (*Sato et al., 2023*), our study uncovers an epigenetic origin. Mechanistically, we demonstrated that ARID1A promoter hypermethylation induced transcriptional silencing, as evidenced by reduced protein expression. This epigenetic inactivation drives persistent activation of PI3K/AKT/mTOR signaling, characterized by elevated phosphorylated AKT levels, which subsequently upregulate PD-L1 expression. Notably, this cascade establishes a novel, epigenetically regulated immune checkpoint axis that links DNA methylation to PD-L1-mediated immune evasion. Pharmacological demethylation was performed using the DNA methyltransferase inhibitor 5-aza-CdR to validate the reversibility of this axis. Treatment with the 5-aza-CdR reduced ARID1A methylation, restored its expression, suppressed PI3K/AKT/mTOR signaling, and downregulated PD-L1 expression. Rescue experiments with the AKT agonist SC79 completely eliminated the 5-aza-CdR-induced effects, confirming that PD-L1 regulation is dependent on the ARID1A-PI3K/AKT/mTOR signaling cascade. The mechanism by which ARID1A promoter hypermethylation activates the PI3K/AKT/mTOR pathway remains incompletely understood. Previous studies indicate that epigenetic silencing of tumor suppressors (*e.g.*, *via* PTEN promoter hypermethylation) is a central driver of aberrant PI3K/AKT activation (*Kang, Lee & Kim, 2002*). As a core subunit of the SWI/SNF chromatin remodeling complex (*Mandal et al., 2022*), ARID1A deficiency may impair chromatin accessibility and lead to transcriptional repression of PTEN and other tumor suppressors. Our preliminary validation supports this hypothesis: analysis of cg05445839 methylation data revealed significantly lower PTEN expression in ARID1A-hypermethylated tumors compared to hypomethylated controls ($p = 0.0005$, Fig. S6K), and treatment of HGC-27 cells with the DNA demethylating agent 5-aza-CdR markedly increased PTEN protein levels concomitant with suppressed p-AKT and PD-L1 expression (Fig. S6L); Future studies employing chromatin immunoprecipitation (ChIP) are warranted to determine whether ARID1A directly binds the PTEN promoter. Collectively, beyond genetic ARID1A lesions, our work establishes ARID1A promoter methylation as a reversible epigenetic switch controlling oncogenic PI3K/AKT signaling. Based on these findings, we propose exploring a triple-combination strategy for *ARID1A-* hypermethylated gastric cancer: 5-aza-CdR to potentially restore tumor suppression and block PD-L1 upregulation, PD-1 inhibitors to counteract immune evasion, and AKT inhibitors (*e.g.*, capivasertib) to mitigate residual pathway activation. AKT inhibitors play a key role by directly blocking AKT phosphorylation, enhancing PD-L1 suppression, and preventing compensatory pathway reactivation—a common resistance mechanism to epigenetic monotherapy (*Cretella et al., 2019*). This synergy ensures sustained PI3K/AKT/mTOR inhibition while linking epigenetic reprogramming to immune potentiation, providing a mechanism-driven solution for overcoming tumors that are resistant to conventional therapies. This preclinical concept provides a rationale for future investigation in advanced

models and early-phase trials. While our study supports this triple therapy for ARID1A-hypermethylated gastric cancer, its clinical use requires careful toxicity management: Blood-related side effects from 5-aza-CdR can be reduced using lower doses (0.1 mg/kg) (*Zhao et al., 2025*). Additional checks during treatment are needed for sugar/heart risks linked to AKT and PD-1 inhibitors (*Huang et al., 2023*; *Ma et al., 2024*). Crucially, giving drugs in order matches their actions—5-aza-CdR resets gene switches, PD-1 drugs activate immune cells, and AKT drugs like capivasertib stop cancer's backup survival plans. Early trials should test this stepwise approach and precise patient selection.

Analysis of ARID1A expression across 33 cancer types revealed robust correlations with four immune modalities: immunosuppressive genes, immunostimulatory genes, chemokines, and chemokine receptors. Emerging evidence indicates that dysregulated DNA methylation in the epigenome modulates the immunogenicity of tumor and immune cells within the TME (*Hogg et al., 2020*). Further investigations demonstrated that ARID1A hypermethylation mediates immune checkpoint inhibition *via* PD-L1 and significantly reshapes the tumor immune microenvironment (TIME). Notably, TIME has been shown to exert greater influence than individual immune checkpoints in governing tumor immune surveillance and escape (*Zhang & Chen, 2016*; *Binnewies et al., 2018*). Integrative analysis using seven algorithms (CIBERSORT, EPIC, *etc.*) identified distinct immune infiltration patterns in hypermethylated tumors, such as elevated pro-tumoral M0 macrophage infiltration and reduced anti-tumor immune cells, including resting memory CD4$^+$ T cells and activated dendritic cells. This pattern aligns with the epigenetic immunomodulatory mechanisms reported by *Ge et al. (2022)* in gliomas, suggesting that ARID1A methylation drives pan-cancer immune evasion by reprogramming macrophage polarization and suppressing T-cell activation. Notably, immune cells within the TIME play a key role in maintaining homeostatic equilibrium (*Bruni, Angell & Galon, 2020*). In GC, the tumor immune score reflects the strength of anti-tumor immunity, while elevated Immune/Stromal/ESTIMATE scores have been established as independent prognostic factors for poorer overall survival (*Zhu et al., 2020*; *Zeng et al., 2022*). In our cohort, ARID1A-hypermethylated STAD tumors exhibited elevated Immune and ESTIMATE scores ($p < 0.05$), mechanistically linking promoter hypermethylation to immunosuppressive TME remodeling.

A notable observation in our study is the co-occurrence of ARID1A promoter hypermethylation and somatic mutations exclusively in the hypermethylated subgroup (25.6% of cases), whereas no such overlap was found in the hypomethylated cohort. This pattern suggests a potential "two-hit" inactivation mechanism in gastric carcinogenesis, wherein the tumor-suppressive function of ARID1A is simultaneously disrupted by epigenetic silencing (*via* promoter hypermethylation) and genetic aberrations. Interestingly, this phenomenon mirrors the findings in ovarian clear cell carcinoma, where ARID1A mutations are associated with promoter hypermethylation and transcriptional repression of downstream targets such as IRX1, TMEM101, and TRIP6 (*Li et al., 2024*). However, our study extends this paradigm by implicating ARID1A as a dual target of epigenetic and genetic alterations in GC. The interplay between ARID1A mutations and DNA hypermethylation may reflect a self-reinforcing loop. ARID1A inactivation (*via* mutation

or methylation) can impair SWI/SNF complex-mediated chromatin remodeling, thereby facilitating aberrant DNA methyltransferase activity (*Ye et al., 2014*; *Yamamoto & Imai, 2015*). Conversely, global hypermethylation may predispose cells to replication errors or defective mismatch repair (*Xiong et al., 2001*), increasing the likelihood of ARID1A somatic mutations (*Shen et al., 2018*). This bidirectional crosstalk could amplify genomic instability, ultimately driving tumor progression in the ARID1A-hypermethylated GC molecular subset.

There exist several limitations to this study. First, while our *in vitro* models established mechanistic causality for ARID1A hypermethylation in driving immune evasion, they cannot fully replicate dynamic tumor-immune interactions. Future work employing patient-derived xenografts (PDX) will be essential to validate these findings in physiological contexts. Second, although multi-omics analyses (TCGA/TISIDB) revealed robust associations between ARID1A methylation and immune phenotypes, prospective validation in immunotherapy-treated cohorts is planned to establish clinical predictive value. Finally, 5-aza-CdR's genome-wide demethylation effects preclude exclusive attribution of immune modulation to ARID1A. We will address this by applying locus-specific methylation editing (dCas9-DNMT3A/TET1) to isolate ARID1A's immunoregulatory functions in future studies.

## CONCLUSION

Collectively, this study delineates a novel epigenetic-immune interplay in gastric carcinogenesis, wherein ARID1A promoter hypermethylation drives immune evasion through PD-L1 upregulation and immunosuppressive microenvironment remodeling. Our findings provide a translational framework: (1) ARID1A methylation status combined with PD-L1 expression may jointly stratify GC patients for ICB therapy, pending clinical validation; (2) 5-aza-CdR enhances PD-1 inhibition efficacy in ARID1A-silenced GC models, suggesting epigenetic priming may overcome immune resistance; and (3) Methylation-guided stratification proposes a biomarker hypothesis for triple combination therapy (5-aza-CdR + ICB + AKT inhibitor). These insights establish the mechanistic basis for biomarker-driven therapeutic development in ARID1A-hypermethylated GC.

### Abbreviations

| | |
|---|---|
| **GC** | Gastric cancer |
| **ICB** | immune checkpoint blockade |
| **TME** | tumor microenvironment |
| **ARID1A** | AT-rich interaction domain 1A |
| **SWI/SNF** | Switch/Sucrose Non-Fermentable |
| **5-aza-CdR** | 5-Aza-2'-deoxycytidine |
| **MeDEGs** | Methylation regulated differentially expressed genes |
| **GO** | Gene Ontology |
| **KEGG** | Kyoto Encyclopedia of Genes and Genomes |
| **BP** | biological processes |
| **MF** | molecular functions |

| CC | cellular components |
|---|---|
| **FDR** | false discovery rate |
| **PPI** | Protein-protein interaction |
| **GSEA** | Gene Set Enrichment Analysis |
| **qRT-PCR** | Quantitative real-time PCR |
| **MSP** | methylation-specific PCR |
| **PBS** | phosphate-buffered saline |
| **SDS-PAGE** | sodium dodecyl sulfate-polyacrylamide gel electrophoresis |
| **IPS** | Immunophenoscore |
| **TCR** | T-cell receptor |

### Funding

This work was supported by the Shandong Provincial Natural Science Foundation (ZR2022MH048), and the Qilu Health Leading Talents Training Project. The funders had no role in study design, data collection and analysis, decision to publish, or preparation of the manuscript.

### Grant Disclosures

The following grant information was disclosed by the authors:
Shandong Provincial Natural Science Foundation: ZR2022MH048.
Qilu Health Leading Talents Training Project.

### Competing Interests

The authors declare there are no competing interests.

### Author Contributions

- Xueqin Duan conceived and designed the experiments, performed the experiments, analyzed the data, prepared figures and/or tables, authored or reviewed drafts of the article, and approved the final draft.
- Xingfa Huo conceived and designed the experiments, performed the experiments, analyzed the data, prepared figures and/or tables, and approved the final draft.
- Yuming Zhang conceived and designed the experiments, analyzed the data, prepared figures and/or tables, and approved the final draft.
- Hongwei Lan analyzed the data, prepared figures and/or tables, and approved the final draft.
- Fangfang Yang conceived and designed the experiments, prepared figures and/or tables, and approved the final draft.
- Xiaochun Zhang conceived and designed the experiments, performed the experiments, analyzed the data, prepared figures and/or tables, authored or reviewed drafts of the article, and approved the final draft.
- Na Zhou conceived and designed the experiments, performed the experiments, analyzed the data, prepared figures and/or tables, authored or reviewed drafts of the article, and approved the final draft.

## Data Availability

The original image data for the western blot, plate clone formation assay, transwell assay, and agarose gel electrophoresis; and the raw qPCR data for mRNA expression (Figures 5G, 5J, 6C, and 6D) are available in the Supplementary Files.

The raw flow cytometry image data (Figures 8M, 9G, and S5M) are available at Figshare: Duan, Xueqin (2025). Original data for Supplementary Figure8M,9G and Figure S5M. figshare. Dataset. Available at https://doi.org/10.6084/m9.figshare.30059362.v1.

## Supplemental Information

Supplemental information for this article can be found online at http://dx.doi.org/10.7717/peerj.20251#supplemental-information.

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
