# Peer review of "Breaking epigenetic shackles: targeting ARID1A methylation and the PI3K/AKT/mTOR-PD-L1 axis to overcome immune escape in gastric cancer"

_PeerJ, doi:10.7717/peerj.20251_

## Round 0.1 · original submission · Major Revisions

· Academic Editor

Major Revisions

The authors are requested to carefully revise the manuscript and answer the questions raised by the reviewers.

Reviewer 1 ·

Basic reporting

The manuscript presents a compelling and timely topic; however, it falls short in several key areas:
- The manuscript suffers from overly complex and dense phrasing throughout, such as in the abstract, results, and discussion. Simplifying technical jargon and improving sentence structure would significantly enhance clarity.
- While the background is generally well-constructed, several claims (such as, ARID1A epigenetic regulation in GC) require citation of more recent literature from 2023–2025 to support novelty claims.
- Figures are abundant but are often crowded with excessive labels and overlapping statistical annotations. Moreover, raw data sharing is incomplete or poorly described in the supplementary files-full datasets including original methylation and RNA expression profiles must be made available and well-annotated.

Experimental design

The research question is novel and aligns with journal scope, but some aspects of the design are underdeveloped:
- While many methods are detailed, critical steps, such as MSP conditions, selection criteria for DEGs, or parameter thresholds in bioinformatic tools, lack replicable precision.
- Only two cell lines (AGS, HGC-27) are used without validation in patient-derived xenografts (PDX) or other biological systems, which weakens translational claims.
- Although this is a bioinformatics and cell-line-based study, an explicit ethics waiver or confirmation of data reuse standards (such as, TCGA licensing) should be stated more clearly.

Validity of the findings

The conclusions are promising but currently overstated relative to the presented data.
- The leap from cell-line demethylation to clinical prediction of ICB response is speculative without in vivo or patient cohort validation.
- While p-values and correlations are noted, statistical correction for multiple testing (such as, FDR in methylation-expression analyses) is inconsistently reported.
- Figures suggest strong effects, but underlying data (such as, full dose-response data, gating strategies for flow cytometry) are not shown or insufficiently described.

Reviewer 2 ·

Basic reporting

This study presents a significant and timely investigation into the epigenetic silencing of ARID1A in gastric cancer (GC) and its functional/immunological consequences. The work integrates multi-omics bioinformatics analyses with robust in vitro functional validation, revealing novel mechanistic insights with clear translational potential. While the findings are compelling and well-supported, some aspects require clarification or further validation to strengthen the manuscript's impact.

Experimental design

The in vitro experiments are well-designed and provide strong evidence:
1 Pharmacological demethylation (5-Aza-CdR) convincingly restores ARID1A expression and suppresses malignant phenotypes (proliferation, invasion, apoptosis resistance).
2 The mechanistic link between ARID1A loss, PI3K/AKT/mTOR pathway activation, and PD-L1 upregulation is clearly established.

Validity of the findings

1 Rescue experiments using the AKT agonist SC79 are particularly compelling, solidifying the proposed ARID1A-PI3K/AKT/mTOR-PD-L1 axis as a key epigenetic-immune cascade.
2 The connection between ARID1A hypermethylation and elevated immune/ESTIMATE scores, combined with the PD-L1 link, provides a strong rationale for proposing ARID1A promoter hypermethylation as a dual biomarker for predicting ICB response and patient stratification. This bridges basic science and clinical application effectively.
3 The proposed Phase II trial concept testing a 5-aza-CdR + ICB + AKT inhibitor combination regimen is a logical and exciting translational extension of the findings.
4 Provide deeper mechanistic insight into the ARID1A -> PI3K/AKT link (if feasible with existing data/resources).
5 Include in vivo data demonstrating the functional consequences of ARID1A demethylation (phenotypic suppression, impact on TME/PD-L1) and preferably, initial proof-of-concept for the combination approach.
6 Strengthen the biomarker claim by analyzing ARID1A methylation status in relation to ICB response in a clinical GC cohort (if accessible).
7 Discuss the practicalities and toxicity risks of the proposed combination regimen more thoroughly.

Reviewer 3 ·

Basic reporting

.

Experimental design

.

Validity of the findings

.

Additional comments

1. The abstract should include objectives, materials, and methods or only methods, results, and conclusions. This part is a little confusing.
2. Write the full name for PI3K/AKT/mTOR in the abstract section.
3. Risk factors for GC in the introduction
4. Short presentation of PI3K/AKT/mTOR dysregulation in GC pathogenesis in the discussion

---

## Round 0.2 · accepted · Accept

· Academic Editor

Accept

After revisions, all reviewers recommend publishing the manuscript. I also reviewed the manuscript and found no obvious risks to publication. Therefore, I also approved the publication of this manuscript.

Reviewer 1 ·

Basic reporting

Please check the comments below.

Experimental design

Please check the comments below.

Validity of the findings

Please check the comments below.

Additional comments

Well-revised; authors addressed all prior concerns.

Reviewer 2 ·

Basic reporting

no comment

Experimental design

no comment

Validity of the findings

no comment

Additional comments

The author has explicitly addressed the issues I raised, and the article now meets the journal's standards.